# CONDITION MATTERS IN FULL-HEAD 3D GANS

**Heyuan Li**[1,†,∗]   **Huimin Zhang**[1,†]   **Yuda Qiu**[1]   **Zhengwentai Sun**[1]   **Keru Zheng**[1]
**Lingteng Qiu**[2,‡]   **Peihao Li**[2]   **Qi Zuo**[2]   **Ce Chen**[1]   **Yujian Zheng**[5]   **Yuming Gu**[6]
**Zilong Dong**[2,✉]   **Xiaoguang Han**[1,3,4,✉]

[1]School of Science and Engineering, The Chinese University of Hong Kong, Shenzhen
[2]Tongyi Lab, Alibaba Group   [3]Shenzhen Future Network of Intelligence Institute
[4]Guangdong Provincial Key Laboratory of Future Networks of Intelligence, The Chinese
University of Hong Kong, Shenzhen   [5]MBZUAI   [6]USC

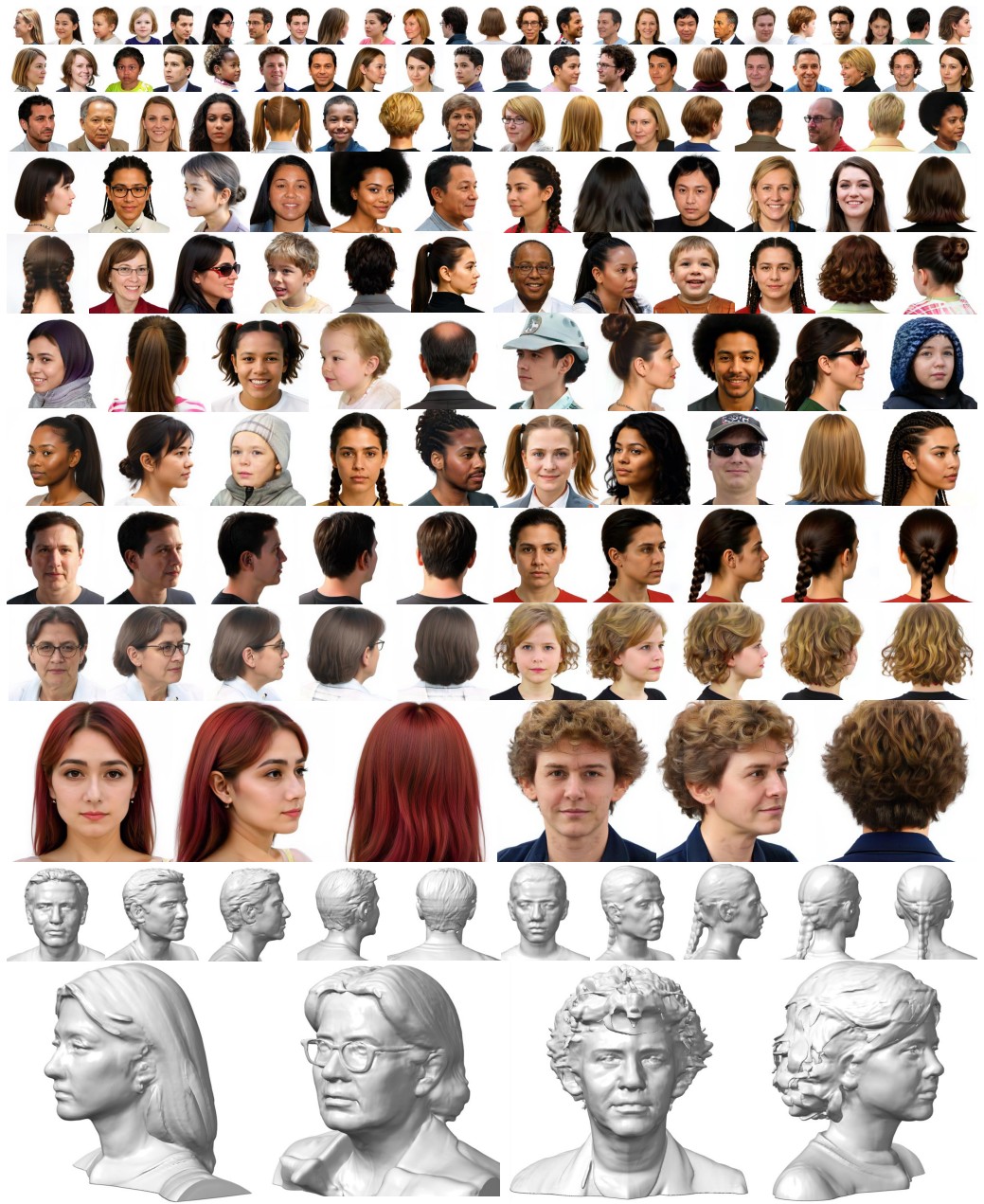

Figure 1: Our full-head model, BalanceHead, is novelly conditioned on view-invariant semantic features to generate high-fidelity and diverse 3D heads. From top to bottom, rows 1–7 show random-view renderings; rows 8–10 display multi-view renderings; rows 11-12 visualize the geometries of the corresponding 3D heads.

---

∗Internship at Tongyi Lab.   †Equal contribution.   ‡Team lead.   ✉Corresponding authors.

## ABSTRACT

Conditioning is crucial for stable training of full-head 3D-aware GANs. Without any conditioning signal, the model suffers from severe mode collapse, making it impractical to training (fig. 2(a,b)). However, a series of previous full-head 3D-aware GANs conventionally choose the view angle as the conditioning input, which leads to a bias in the learned 3D full-head space along the conditional view direction. This is evident in the significant differences in generation quality and diversity between the conditional view and non-conditional views of the generated 3D heads, resulting in global incoherence across different head regions (fig. 2(d-i)). In this work, we propose to use *view-invariant semantic feature* as the conditioning input, thereby decoupling the generative capability of 3D heads from the viewing direction. To construct a view-invariant semantic condition for each training image, we create a novel synthesized head image dataset. We leverage FLUX.1 Kontext to extend existing high-quality frontal face datasets to a wide range of view angles. The image clip feature extracted from the frontal view is then used as a shared semantic condition across all views in the extended images, ensuring semantic alignment while eliminating directional bias. This also allows supervision from different views of the same subject to be consolidated under a shared semantic condition, which accelerates training (fig. 2(c)) and enhances the global coherence of the generated 3D heads (fig. 1). Moreover, as GANs often experience slower improvements in diversity once the generator learns a few modes that successfully fool the discriminator, our semantic conditioning encourages the generator to follow the true semantic distribution, thereby promoting continuous learning and diverse generation. Extensive experiments on full-head synthesis and single-view GAN inversion demonstrate that our method achieves significantly higher fidelity, diversity, and generalizability. Project page: https://lhyfst.github.io/balancehead/.

## 1 INTRODUCTION

Generalizable 3D head synthesis has long been a central topic in computer vision and graphics, driven by its broad applications in digital entertainment, virtual avatars for augmented and virtual reality, and next-generation content creation pipelines. Traditional approaches, such as 3DMM Blanz & Vetter (1999) and FLAME Li et al. (2017), rely on large-scale, expensive 3D data sources like depth scans or multi-view image collection. These methods, however, are often limited in capturing photorealistic facial details and back-head region with diverse hairstyles.

3D-aware GANs have been widely applied to 3D head generation. Early approaches used unconditional 3D-aware GANs Nguyen-Phuoc et al. (2019); Chan et al. (2021b) on near-front-view images (e.g., FFHQ), which did not report significant issues, as the view is relatively fixed and thus easier to learn. However, EG3D Chan et al. (2022) demonstrated that introducing a view condition helps disentangle view-dependent information from other features, facilitating the learning of 3D priors and leading to noticeable quantitative improvements. Nevertheless, it also identified that using view-conditional generation can lead to the 2D billboard artifacts. To address this, EG3D proposed a regularization method based on randomly swapping the conditioning view. While this approach provides some improvement in near-front views, it does not fully resolve the issue.

Subsequent full-head 3D-aware GANs An et al. (2023); Li et al. (2024); Wu et al. (2024); He et al. (2025) largely build upon EG3D: they adopt and improve the tri-plane-like representation, while inheriting the view-conditioning strategy and its associated regularization. However, when trained on data covering a much broader range of views, especially full 360° perspectives, these models exhibit numerous artifacts and strong conditional view dependency. Specifically, the generation quality and diversity at the conditional view are significantly higher than at other views, resulting in clear global inconsistencies (Fig. 2(d,e)). To ensure that the front face appears natural, which is much more sensitive to human observation than other head regions, all these methods adopt a common trick from EG3D. Specifically, they use conditioning signals for all view angles during training, but only employ a fixed frontal view at inference time. This approach prioritizes high-quality and diverse outputs in the front-facing region at the cost of limited representational capacity for rear views. As a result, the back-head region often struggles with limited hairstyle variation and

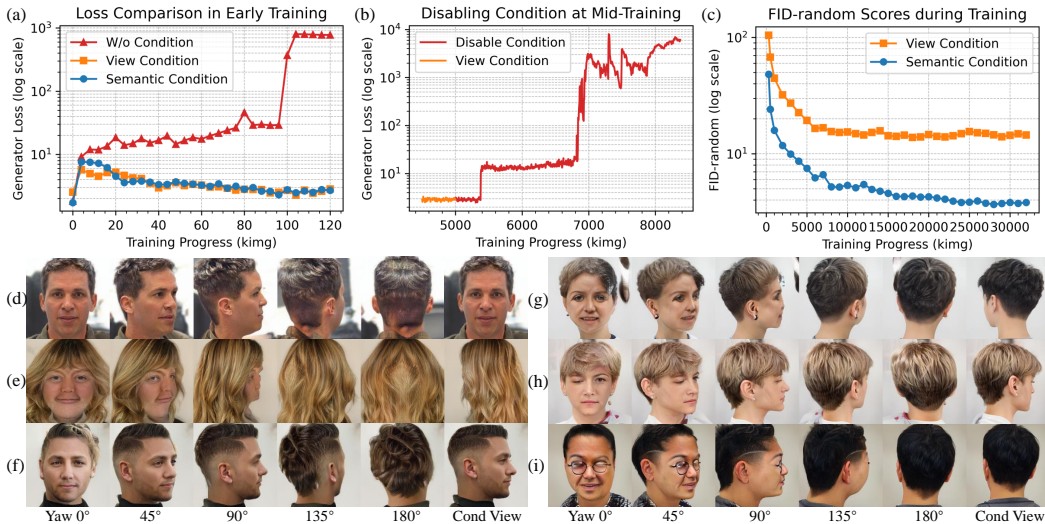

Figure 2: (a) No conditioning leads to early mode collapse and unstable training. (b) Disabling view conditioning mid-training causes rapid collapse within 1000 kimg. (c) Semantic conditioning enables faster and more effective training. (d–i) View-conditioned models show strong directional bias and global incoherence; while conditional views are realistic, non-conditional views are distorted and inconsistent. (d,e), (f,g), and (h,i) are results conditionally generated by random conditional views from PanoHead An et al. (2023), SphereHead Li et al. (2024), and HyPlaneHead Li et al. (2025).

is prone to distortions and artifacts. This is one of the reasons why SphereHead Li et al. (2024) reports frequent artifacts in this area. Moreover, such an inference setting limits the overall diversity of the generated output, as it only utilizes a small subset of the conditions available during training.

To avoid the limitations of view conditioning, we first consider removing it entirely. However, extensive experiments show that omitting the condition in the early training stage leads to rapid mode collapse and makes training infeasible. Alternatively, we increase the view-swap probability to 1 during the middle phase of training, which is equivalent to disabling the condition. This setting typically leads to mode collapse within 1000k images, as shown in Fig. 2(a,b). Therefore, for full-head generation, conditioning is essential, but a view-invariant conditioning strategy is required.

In this paper, we propose using *front-view semantic feature* as the conditioning input. Specifically, for a person's multi-view images, we align the conditioning of all views to the semantic feature of the frontal view, since it contains the most comprehensive information, including the essential facial features and the description of the hairstyle. By unifying all views under the same condition, the dependency on specific viewing directions is effectively removed, making the conditioning view-invariant. However, such an approach heavily relies on multi-view data, which are rare and expensive to scale up.

Instead, we leverage the strong generative capability of 2D image generation models, i.e. Flux.1 Kontext Batifol et al. (2025), to expand high-quality real-world front head images into multi-view collections, thereby obtaining the required multi-view data. While 2D generation models cannot guarantee full 3D consistency, they excel at preserving the global semantic content, such as identity and clothing appearance. In our training process, these data are treated as single-view inputs, so incomplete 3D consistency does not affect learning; instead, the model learns to ensure view consistency through adversarial training and a tri-plane-like representation. Moreover, this data generation strategy significantly alleviates the limitations of in-the-wild datasets used in previous full-head models, which are difficult to collect and often suffer from imbalanced distributions of image quality, quantity, and diversity across different views. Such imbalances lead to problematic supervision during training. In contrast, our approach guarantees that both data and semantic condition follow consistent distributions across all views, greatly benefiting the training of full-head models.

In addition, using semantic conditioning offers another benefit: it enables the use of large-scale data while maintaining diversity. The nature of adversarial learning often causes the generator to stall

once it learns a few modes or patterns that successfully fool the discriminator, rather than continuing to explore more diverse outputs. However, our semantic conditioning enforces the generator to produce images consistent with randomly sampled semantic conditions from the dataset, thereby strictly adhering to the real data distribution. Moreover, as the amount of data increases, the supervision in the condition space becomes denser, leading to better model performance. As a result, the model is able to continuously improve as the training data scale up.

Therefore, we construct a *10-million-level* 360° full-head image dataset using the aforementioned data generation method. We train our semantic-conditional 3D-aware GAN on this dataset, resulting in a powerful 3D full-head model that achieves high diversity and fidelity with significantly reduced artifacts. Extensive experiments demonstrate its state-of-the-art performance.

In summary, our work makes the following major contributions:

- We conduct the first in-depth analysis of the limitations of view-conditioning in full-head 3D-aware GANs and propose a novel semantic-conditional 3D-aware GAN to address these issues.
- To facilitate semantic-conditional training, we construct a synthetic dataset with balanced image quality, quantity, and diversity across views, and with uniformly distributed semantic conditions.
- Our method achieves state-of-the-art performance on both full-head synthesis and single-view 3D GAN inversion tasks, demonstrating the effectiveness of our proposed data and model design.

## 2 RELATED WORK

**3D-Aware GANs**  3D-aware Generative Adversarial Networks Szabó et al. (2019); Shi et al. (2021); Liao et al. (2020); Gadelha et al. (2017); Nguyen-Phuoc et al. (2020; 2019); Deng et al. (2022); Or-El et al. (2022); Chan et al. (2021a); Schwarz et al. (2020) aims to synthesize view-consistent images from diverse viewpoints by learning a scene's underlying 3D geometry and appearance from a collection of 2D images. Among them, Gram Deng et al. (2022) proposed to constrain the radiance field learning on 2D manifolds, embodied as a set of learnable implicit surfaces. EG3D Chan et al. (2022) introduced an efficient hybrid representation that projects a tri-plane onto a 3D volume, a technique that has since been widely adopted. GGHead Kirschstein et al. (2024) replaced the NeRF-based representation with 3D Gaussian Splatting (3DGS) by parameterizing 3D Gaussian heads as UV maps, which enables efficient training on 2D convolutional neural networks (CNNs). While these methods achieve impressive visual quality, they still face significant challenges, particularly in synthesizing head images from large-angle views.

**360° Full-Head Generation**  To address the limitations of narrow-view synthesis and enable full-head generation, researchers have focused on extending existing architectures. PanoHead An et al. (2023) took a significant step by constructing a in-house training dataset with large view angles and expanding the tri-plane's capacity with parallel feature planes. SphereHead Li et al. (2024) improved upon this by introducing a spherical coordinate representation and a novel view-image consistency loss, effectively mitigating artifacts such as mirroring and multiple faces. HyPlane-Head Li et al. (2025) further introduces a hybrid planar-spherical representation to reduce artifacts while maintaining high-quality detail rendering. More recently, 3DGH He et al. (2025) introduced a method that models the head and hair regions separately, reducing the large distribution divergence between front and back views. Alternatively, DiffPortrait360 Gu et al. (2025) and AvatarBack Xin et al. (2025) directly trained a back-head prior to synthesize the back-view image as inference via 2D diffusion or 2D GAN, alleviating the burden of 3D generator.

Despite these architectural and methodological improvements, artifacts persist and diversity is limited, especially in the back view. This is primarily due to the lack of high-quality back-view images and balanced data distribution in existing training sets. SOAP Liao et al. (2025) attempted to tackle this issue by collecting a large-scale 3D head dataset with 24k models and rendered training images from full range views. However, scaling up the number of unique identities with this approach remains a significant challenge. ID-Sculpt Hao et al. (2024)and Portrait3D Wu et al. (2024) attempted to refine the 3D head inverted from a 3D-aware GAN by Score Distillation Sampling, thereby avoiding dependency on training data, but this leads to color oversaturation and loss of fine details.

**Human Head Portrait Datasets**  Generative models require large-scale, high-quality portrait datasets to ensure diverse and realistic outputs. While widely used datasets like FFHQ Karras et al.

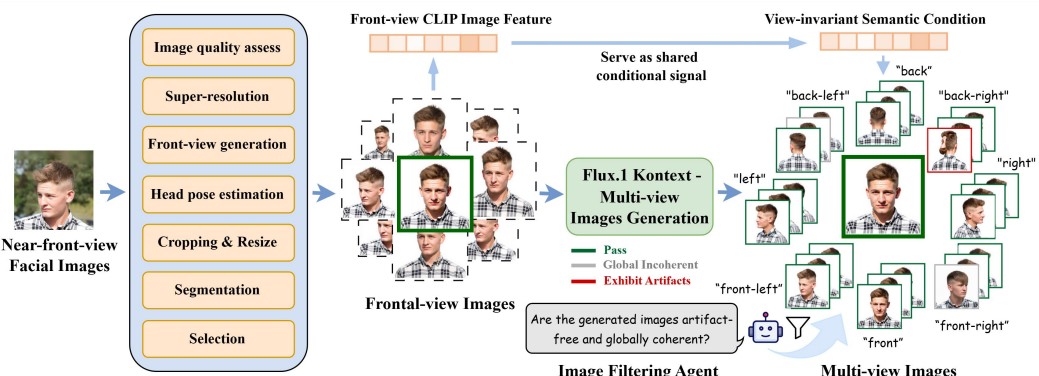

Figure 3: Overview of our data generation pipeline. The first stage selects a large number of near-front-view head and facial images, generates their corresponding front-view images using Flux.1 Kontext, and performs data preprocessing. In the second stage, we similarly leverage Flux.1 Kontext with different view-angle prompts to extend the front-view images into multi-view collections. Finally, an image filtering agent based on Qwen2.5-VL is employed to remove images with artifacts or global incoherence.

(2018) and CelebA Liu et al. (2015) are available, they mainly contain near-frontal views and are insufficient for full-head generation. The LPFF dataset Wu et al. (2023a) offers a broader view range but still lacks sufficient back-view images. The WildHead dataset Li et al. (2024) includes more back-view samples but suffers from uneven view distribution. This is largely due to natural variations in lighting, appearance, posture, and hairstyle. For example, certain hairstyles, such as Russian braids, are often only captured from the rear. Such imbalances are inevitable in real-world data.

## 3 BALANCEHEAD360 DATASET

### 3.1 DATA GENERATION PIPELINE

As shown in fig. 3, our data generation pipeline consists of two stages. First, we collect approximately 350k high-quality near-front-view facial images from public datasets such as FFHQ Kazemi & Sullivan (2014), CelebA Liu et al. (2015), WildHead Li et al. (2024), and FaceCaption Dai et al. (2024). After preprocessing, we use Flux.1 Kontext Batifol et al. (2025) to generate a frontal-view counterpart for each image. In the second stage, the same model is used to expand the frontal view into multiple perspectives using different view prompts. The CLIP feature extracted from the generated front-view image serves as the shared conditional label for all extended multi-view images. For detailed technical operations, including the hyperparameters and specific prompts used for all models presented in this section, please refer to the supplementary material.

**Front View Images Synthesis** For each input near-front-view real image, we first use Hyper-IQA Su et al. (2020) to estimate image quality. Images with a score above 60 are retained, while those scoring below 35 are directly discarded. Images falling between these thresholds are further enhanced using a super-resolution model HYPIR Lin et al. (2025). We then employ Flux.1 Kontext Batifol et al. (2025) to generate multiple frontal-view images of the same subject while preserving their appearance and removing the background and non-central persons. We use VG-GHeads Kupyn et al. (2024) to estimate head pose and crop the image to center-align the head. We then use DAViD Saleh et al. (2025) to obtain the mask of the foreground person. Next, we filter out images with an absolute yaw angle greater than $10°$. For the remaining ones, we extract identity features using ArcFace Deng et al. (2019) and select the image with the smallest cosine similarity to the original as the synthesized front view image.

**Multi-view Images Synthesis** In this stage, we leverage the strong viewpoint control and identity preservation of Flux.1 Kontext. Given a front-view image generated in the previous stage, we provide different view prompts to generate images from various perspectives while preserving the

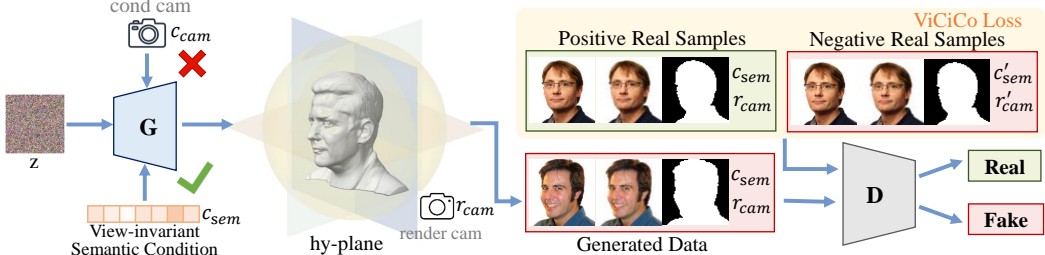

Figure 4: The Overview of our BalanceHead pipeline.

subject's appearance. However, due to the inherent instability of 2D generative models, the output occasionally contains artifacts or fails to maintain consistent appearance across views. To filter out such problematic images, we employ an image filtering agent based on Qwen2.5-VL Bai et al. (2025). This agent effectively removes corrupted images and reduces cases where the subject's appearance changes significantly. Finally, we perform head pose estimation, cropping, and mask extraction on the images using the methods described above.

## 3.2 ROBUSTNESS OF 3D-AWARE GANS TO IMPERFECT MULTI-VIEW DATA

It is important to note that although the generated multi-view data do not strictly guarantee 3D consistency, we find this does not hinder training. First, during training, we treat each view as an independent input without enforcing explicit multi-view supervision. Second, 3D-aware GANs learn through adversarial training and use a tri-plane-like representation to encode 3D structures. As a result, most 2D artifacts become hard examples for the generator, as they are difficult to reproduce using geometric representations in the tri-planes. Consequently, the generator naturally avoids learning these outlier samples. Therefore, 3D-aware GANs are well-suited for leveraging data from 2D models, benefiting from their high quality, controllability, and strong appearance consistency, while remaining robust to occasional artifacts and preserving 3D coherence.

## 3.3 BALANCEHEAD360 DATASET OVERVIEW

Via the aforementioned data generation process, we construct a synthetic dataset containing 11.2 million 360° full-view head images with condition labels. The entire data generation and processing pipeline was executed on a computing cluster with 400 Nvidia A10 GPUs over a period of 26 days.

We name this dataset BalanceHead360, as it addresses the imbalance in real-world data used by prior works. In our dataset, the distribution of image quantity, quality, and diversity is balanced across all viewing directions, significantly alleviating the supervision bias observed in SphereHead. Moreover, each subject shares a consistent semantic condition across views, ensuring that the distribution of each condition remains uniform across all perspectives. The inclusion of full-view images for every condition enables effective training with 3D-aware GANs.

Additionally, our data generation pipeline leverages a large number of near-front-view real images and extends them to multiple views, resulting in generated images that follow a natural distribution. Despite being synthetic, the dataset avoids imposing an artificial distribution, which enhances its generalization to real-world data. This design also enables significant scalability by making use of the abundant existing frontal head images.

## 4 SEMANTIC-CONDITIONAL 3D-AWARE GANS

### 4.1 DEFINITION

Unlike previous unconditional 3D-aware GANs and those conditioned on view angles, we propose a new class of 3D-aware GANs that use view-invariant semantic features as conditions. This design eliminates view-dependent cues from the conditioning signal, thereby breaking the correlation between generation capability and specific views and effectively addressing the prevalent directional bias in prior methods. Furthermore, it ensures that the generator learns to produce diverse outputs

by aligning with the true semantic distribution of the data. By consolidating supervision across different views under a shared semantic condition, the model enforces semantic consistency across all perspectives for each generated sample, which enhances global coherence and improves training efficiency. We refer to this class of models as *semantic-conditional 3D-aware GANs*.

## 4.2 VIEW-INVARIANT SEMANTIC FEATURE

As previously analyzed, using view as a condition can introduce directional bias in the 3D heads generated by 3D-aware GANs. Therefore, we aim to derive a view-invariant semantic feature from the images to serve as a new conditioning signal. Ideally, one could align the semantic features of all views of a subject, after removing their respective view-specific information, to a central representation that is shared across all perspectives. However, in practice, it is highly challenging to eliminate view-specific content from each image, as different views contain distinct visual information: for example, frontal views capture rich facial features but lack hair details, while rear views emphasize hair but miss facial features.

Since it is infeasible to remove view-specific information from individual images, we instead anchor all views to a single reference image, allowing them to share a common condition and thus decoupling the condition from any specific viewpoint. Although no single view captures 100% of the full-head semantic information, the frontal view provides the most comprehensive signal among all viewpoints. It fully captures the facial region, which is most critical to human perception, and also conveys substantial global cues about hairstyle, hair color, clothing, and overall appearance. Therefore, we choose the CLIP image feature of the front view as the shared view-invariant semantic information across all views, thereby avoiding directional bias in the generation process.

## 4.3 VIEW-IMAGE AND CONDITION-IMAGE CONSISTENCY LOSS (VICICO LOSS)

Counterintuitively, despite our model being trained on view-balanced data, we occasionally observed multiple-face artifacts in the back head region, which are reported by SphereHead. However, unlike in SphereHead where such artifacts appear early in training, they typically emerge only at later stages in our experiments, suggesting a different underlying cause. Through visualization analysis of checkpoints before and after the artifact occurrence, we found that these artifacts likely arise when the generator struggles to learn complex hairstyles, accessories, or hats due to their high diversity and large geometric and color variations. In this process, the generator may degenerate into generating meaningless floating color patches. As face-like patterns are easier for the discriminator to accept, the model eventually generates facial features within these patches, leading to the multiple-face artifacts.

Fortunately, our model remains compatible with the ViCo loss proposed by SphereHead, which enforces consistency between image content and view information through the discriminator. Although this involves view-related signals, the used views are randomly rendered and independent of the semantic condition, thus avoiding directional bias. Building upon this idea, we construct negative pairs based on view-invariant semantic conditions, enabling the discriminator to jointly enforce consistency between image content and the semantic condition. This further strengthens the alignment between generated images and the true semantic distribution, enhancing both fidelity and diversity. We combine this with the ViCo loss to form a new loss function, View-image and Condition-image Consistency Loss (ViCiCo Loss), which is formulated as follows:

$$\mathcal{L}_{\text{ViCiCo}} = \log(1 - D((I^+, I, I^m), (r'_{\text{cam}}, c'_{\text{sem}}))). \tag{1}$$

As illustrated in fig. 4, for a batch of real data $\{(I^+, I, I^m), (r_{\text{cam}}, c_{\text{sem}})\}$, we randomly shuffle either the camera label $r_{\text{cam}}$, the semantic condition $c_{\text{sem}}$, or both to form a shuffled pair $(r'_{\text{cam}}, c'_{\text{sem}})$. The resulting negative pairs $\{(I^+, I, I^m), (r'_{\text{cam}}, c'_{\text{sem}})\}$ are then fed into the discriminator $D$.

## 4.4 BALANCEHEAD

We integrate the above technical innovations into a semantic-conditional full-head 3D-aware GAN pipeline, BalanceHead, as shown in fig. 4. Building upon recent state-of-the-art HyPlaneHead model, our generator employs StyleGAN2 as the backbone and uses a hy-plane representation to

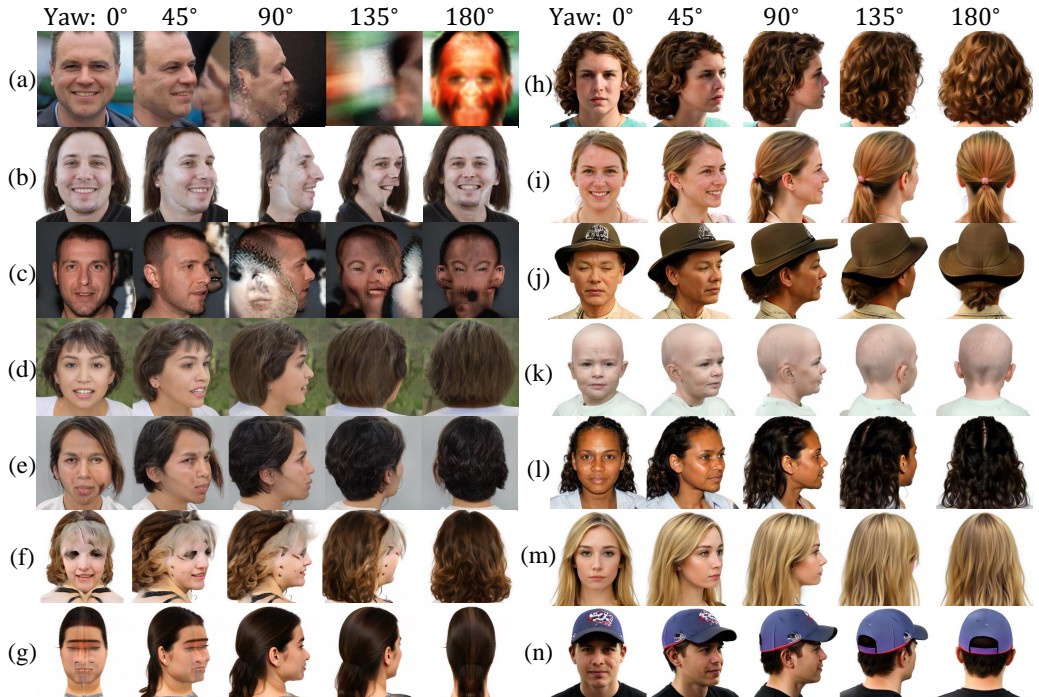

Figure 5: Qualitative comparison with state-of-the art methods. Conditioned on front-view: (a) EG3D (b) GGHead (c) PanoHead (d) SphereHead. (e) HyPlaneHead conditioned on back-view. (f) Our view-conditional baseline conditioned on back-view. (g) Our view-semantic-conditional baseline conditioned on side-view. (h-n) Our BalanceHead conditioned on view-invariant semantic condition.

encode 3D head geometry. The pipeline first renders low-resolution images $I$ via volume rendering and then applies a super-resolution module to generate high-resolution images $I^+$ along with corresponding masks $I^{m+}$. In contrast to previous methods, we condition the generator using a view-invariant semantic feature $c_{\text{sem}}$ instead of camera view information $c_{\text{cam}}$, and incorporate the proposed ViCiCo loss to suppress multiple-face artifacts and ensure consistency between the generated output and the semantic condition. All these design choices collectively enable BalanceHead to achieve high-quality, diverse, and generalizable full-head generation with minimal artifacts.

## 5 EXPERIMENTS

The model was trained on eight NVIDIA H20 GPUs using a batch size of 32. Both the training and synthesized images have a resolution of $512 \times 512$. The training process, which lasted 10 days, exposed the model to a total of 32 million images from our BalanceHead360 dataset.

### 5.1 360° FULL-HEAD SYNTHESIS

**Qualitative Results** We compare our BalanceHead with existing 3D-aware head GANs, including EG3D Chan et al. (2022), GGHead Kirschstein et al. (2024), PanoHead An et al. (2023), Sphere-Head Li et al. (2024), and HyPlaneHead Li et al. (2025). We also evaluate two ablation baselines that share the same training settings as BalanceHead but differ in conditioning strategies: a view-conditional baseline that conventionally uses the view angle as condition, equivalent to Hy-PlaneHead trained on our dataset; and a view-semantic-conditional baseline that uses the image clip feature of the current view instead of the corresponding front view as the conditioning signal. As shown in fig. 5, (a) EG3D and (b) GGHead fail to generate the back-head region, as they are trained on near-front-view datasets and lack suitable 3D representations for non-frontal areas. (c) PanoHead frequently exhibits severe artifacts caused by feature penetration resulting from Cartesian coordinate projection. While (d) SphereHead addresses this issue with a spherical tri-plane representation, its

Table 1: Quantitative results

| Condition | ViCiCo | FID-view ↓ | FID-random ↓ | FID-front ↓ |
|-----------|--------|-----------|--------------|-------------|
| view | - | 9.67 | 13.82 | 8.42 |
| view-semantic | - | 8.63 | 46.24 | 5.90 |
| semantic | - | - | 4.45 | 4.11 |
| semantic | ✓ | - | **3.67** | **3.51** |

suboptimal design limits the generator's capacity, resulting in blurry outputs with poor detail. (e) The recent SOTA method, HyPlaneHead, introduces a hybrid plane representation combining the strengths of tri-plane and spherical tri-plane structures, achieving high-quality results. However, as it still conditions on view angles like previous methods, it can only stably generate realistic samples under front-view conditions. When non-front views are used as input, artifacts and distortions appear in non-conditional directions, limiting its diversity.

We train (f) HyPlaneHead on our newly created 360°-balanced dataset but still observe view-biased performance. To address this, we explore using the image clip feature of each view as a semantic condition. However, since these features inherently contain view-related information that is difficult to disentangle, the resulting outputs continue to exhibit directional bias.

In contrast, our BalanceHead aligns all views to the image CLIP feature of their front view as the conditioning signal, effectively eliminating the haunted directional bias. Combined with our large-scale, 360°-balanced dataset, BalanceHead360, our method achieves stable, high-quality, and globally coherent outputs across all viewing angles for the first time. The learned 3D full-head space not only captures variations in facial expressions and attributes but also spans diverse identities including race, hairstyle, accessories, clothing, and headwear, demonstrating its potential for downstream tasks such as 3D hair modeling and full-head 3D talking avatars.

**Quantitative Results**   We use three variants of the Fréchet Inception Distance (FID) score to quantitatively compare our method with baselines. FID-view evaluates only the conditional view, which is equivalent to the standard FID used in prior works. However, it omits the quality and diversity of non-front views. To address this limitation, HyPlaneHead proposes FID-random, which renders images from random views to evaluate the overall 360° synthesis quality and diversity. This metric is most relevant to our task. FID-front measures outputs conditioned on and rendered from the front view. In this setting, both view-conditional and semantic-conditional methods consistently produce stable, high-quality front-facing portraits, thus reflecting the identity diversity of the full-head space.

As shown in table 1, using view or view-semantic as conditioning leads to significant directional bias, evidenced by the much higher FID-random compared to FID-view. In contrast, semantic conditioning achieves better quality and diversity, reflected in a significantly lower FID-random. The ViCiCo loss further improves consistency between the output and the condition, resulting in even lower FID-random. Even when evaluated only on the front view, our semantic conditioning combined with ViCiCo loss still demonstrates improved diversity.

## 5.2   SINGLE-VIEW GAN INVERSION

We compare single-view GAN inversion results with previous full-head 3D-aware GANs in fig. 6. Specifically, we perform single-view GAN inversion using Pivotal Tuning Inversion (PTI) Roich et al. (2022), which optimizes both the latent code $w$ and generator parameters to match the target image via a combination of pixel-wise L2 loss and image-level LPIPS loss. PanoHead exhibits persistent face-like artifacts on the back head region, while SphereHead generates fewer artifacts but suffers from blurry outputs and lacks fine details such as hair tips and glasses geometry. HyPlaneHead produces better results overall but still struggles with uncommon hairstyles; for example, the unusual hair tips from the second row input are learned as background elements and do not rotate consistently with the head. All these methods tend to produce similar back head regions, primarily due to their imbalanced capability in capturing quality and diversity across views, resulting in a lack of global coherence. In contrast, thanks to conditioning on a view-invariant semantic feature, our BalanceHead faithfully reconstructs 360° full-head images, appropriately filling in invisible back head regions and maintaining global coherence even for outliers.

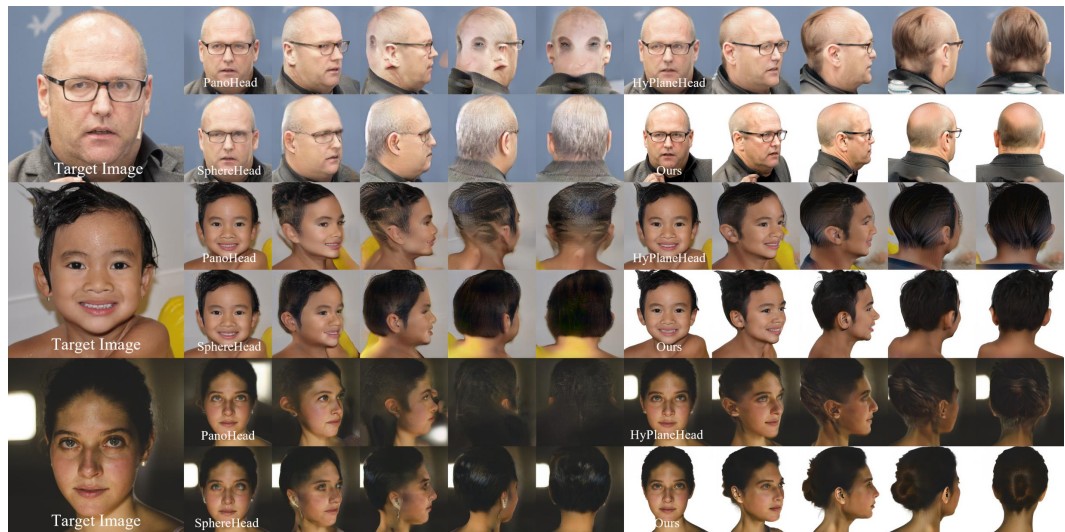

Figure 6: Single-view 3D-aware GAN Inversion.

## 6 DISCUSSION

Trained on large-scale, high-quality images, our proposed 3D full-head model exhibits significant fidelity, diversity, and generalizability. These attributes position the model not only as a potential foundation for specialized applications, such as 3D talking heads and head editing, but also as a high-fidelity 2D/3D data generator capable of supporting downstream tasks like 3D head reconstruction.

Beyond its immediate applications, this work offers a new perspective on the synergy between 2D generative priors and 3D consistency. Specifically, our results demonstrate that fully 3D-consistent representations can be effectively supervised by imperfect, 3D-inconsistent multi-view data. This suggests a potential paradigm shift: rather than relying exclusively on strictly consistent (but expensive) multi-view data or purely single-view data (which often suffers from view imbalance), researchers can leverage the vast, though slightly inconsistent, output of powerful 2D generators. A key takeaway is that future research should perhaps place greater emphasis on developing inconsistency-tolerant 3D training strategies and robust semantic-conditioned models, rather than focusing solely on the pursuit of perfect consistency in the input data. Our method serves as an initial exploration in this direction, highlighting the critical role of view-invariant semantic information in bridging the gap between 2D priors and 3D consistency.

## 7 CONCLUSION

In this work, we address the limitations of view-conditioning in full-head 3D-aware GANs by introducing a novel semantic-conditional approach. Our method enables view-invariant generation through front-view-aligned semantic features and a large-scale, balanced synthetic dataset. Extensive experiments demonstrate that our model achieves superior quality, diversity, and global coherence compared to existing methods.

## 8 ACKNOWLEDGEMENTS

The work was supported in part by Guangdong S&T Programme with Grant No. 2024B0101030002, the Basic Research Project No. HZQB-KCZYZ-2021067 of Hetao Shenzhen-HK S&T Cooperation Zone, by Guangdong Provincial Outstanding Youth Fund with No. 2023B1515020055, the Shenzhen Outstanding Talents Training Fund 202002, the NSFC with Grant No. 62293482, the Guangdong Research Projects No. 2017ZT07X152 and No. 2019CX01X104, the Guangdong Provincial Key Laboratory of Future Networks of Intelligence (Grant No. 2022B1212010001), and the Shenzhen Key Laboratory of Big Data and Artificial Intelligence (Grant No. SYSPG20241211173853027), the Guangdong Province Radio Science Data Center.

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

# A APPENDIX

## A.1 USE OF LLMS

We acknowledge the use of general-purpose Large Language Models (LLMs), such as OpenAI's ChatGPT, as a tool for refining the manuscript. Their role was confined to improving grammatical accuracy, stylistic flow, and clarity. The core intellectual contributions, including the research problem, the design of the BalanceHead framework, and the ViCiCo loss, were developed and executed solely by the human authors. This manuscript's scientific narrative, structure, and all claims are the original work of the authors.

## A.2 DATA GENERATION PIPELINE DETAILS

We detail the configuration of the data generation pipeline in this section.

For both synthesis processes, we set the Classifier-Free Guidance (CFG) scale to 1.0 and the denoising steps to 20. To ensure the quality of the generated images, we designed specific positive and negative prompts, as illustrated in Fig.7.

| Positive Prompt | Negative Prompt |
|---|---|
| A 4k HD candid photo of a person [view], centered composition, remove other people, pure white background, natural shoulder posture, only one person visible, no other people, no obstructions, highly realistic hairstyle, subtle and highly detailed hair strands, photorealistic, keep original features intact, maintain identity, maintain expression, enhance this person using super-resolution, natural skin, low saturation, no retouching, photographed with a DSLR, natural lighting, unstylized, high-resolution, true-to-life, studio-style portrait, remove unnatural edges caused by mirror padding, replicate padding, and edge padding. | obstructions, occlusions, split image, a collage, multiple people, extra people, occluded head, head covered, object in front of face, hand, microphone, acne, pimples, bad skin, oily skin, zits, blackheads, skin blemishes, cluttered background, busy background, text, logo, watermark, low resolution, bad composition, messy background, poor lighting, multiple views, three views, two views, bad anatomy, distorted face, blurred face, out of focus, monochrome, artifacts, noise, blurry, disfigured, deformed, extra limbs, text, signature, unnatural edges, mirror padding, replicate padding, edge padding. |

| [view] | "facing forward, front view", "left profile, left side view", "right profile, right side view", "left-rear view, back-left", "right-rear view, back-right" | "three-quarter view, 45-degree angle, partial side, yaw=45 degree", "side view, lateral view, profile view, yaw=90 degree", "back-side view, rear-left, yaw=135 degree", "back-side view, rear-right, yaw=135 degree", "back view, rear view, posterior view, yaw=180 degree", |
|---|---|---|

Figure 7: Prompts in FLUX.1 Kontext. We use different key words in [view] (bottom in the figure) to control the pose of synthesis head.

The image synthesis process, particularly for multi-view image synthesis, may suffer from identity distortion and visual artifacts. To address this, we developed a two-stage image filtering process to clean the synthesized head images, leveraging Qwen2.5-VL, a comprehensive visual-language model (VLM).

In general, we classify the synthetic head images into two categories based on the complexity of head attributes, including hair, clothing, and accessories. For each generated multi-view image, we compare it with the corresponding front-view image to decide whether to retain or discard it.

In the first stage, we focus on categorizing the images. We define "Standard Images" and "Non-Standard Images" as shown in Fig.8.

In the second stage, we determine whether the synthesized image contains severe artifacts or discrepancies that could indicate image manipulation or inauthenticity. First, we designed prompts for detecting severe artifacts, such as severe distortion of facial features, visible blending/merging errors between the face and background, and more. Then, we developed specific prompts for Standard and Non-Standard images respectively (see Fig.10) to verify whether the synthetic multi-view image shares identical features with the conditional front-view image.

---

**Classification Prompt**

Please determine whether this image falls under 'Standard Image' or 'Non-Standard Image' based on the following criteria.

Standard Image Characteristics:
Minimal head rotation (within ±15°), natural expression, consistent facial feature alignment, natural hair flow, clear facial details, high image quality, and good facial symmetry.
The head should not show noticeable tilt or excessive rotation. Facial features should be aligned consistently across images, with no asymmetry in the face or hairstyle. The image should be well-lit, and the facial features should be clear without obstruction. For example, a person smiling with a natural posture, face aligned, and hair flowing naturally without distortion or misalignment due to head tilt would qualify as a standard image.
In terms of accessories, if items such as glasses, earrings, or hats are present, they should not dominate the image or obscure the face. If the accessories occupy a large portion of the image, such as covering the eyes or ears significantly, the image should be classified as 'Non-standard Image'. Accessories should be a secondary focus, not taking away from the clarity and alignment of the face and head in the image.

Non-standard Image Characteristics:
Significant head rotation (exceeding ±30°), pronounced facial expression changes, misaligned or asymmetrical facial features, hairstyle inconsistent with head angle, obscured or incomplete facial details, poor image quality, and poor facial symmetry.
Images with exaggerated head tilts or rotations, which result in misaligned facial features, exaggerated expressions, or hair misalignment due to head movement, should be classified as non-standard. Also, if the image quality is poor or facial details are hidden or unclear, it should be categorized as non-standard.
Additionally, if accessories like glasses, earrings, or hats occupy a substantial part of the image, such that they obscure the face or distort the natural appearance, the image should be classified as 'Non-standard Image'. For example, if the glasses are large and cover the eyes or the hat is too big and hides part of the face, this should be considered a significant inconsistency.

Please respond with only: 'Standard Image' or 'Non-standard Image', and followed by one concise sentence explaining the key reason.

Figure 8: Classification Prompts in Qwen2.5-VL.

---

**Artifact Perception Prompt**

Please determine whether this image contains severe artifacts or discrepancies that may indicate image manipulation or inauthenticity.

Severe Artifacts Criteria:
If the image shows any unnatural distortions such as:
Severe distortion of facial features or proportions, such as faces with unnatural distortions, abnormal rotations, or disproportionate facial features. This includes instances where heads appear unusually large or small compared to the body.
Inconsistent textures or surface abnormalities, such as overly smooth, overly sharp areas, or unnatural pixelation that suggests image manipulation.
Unnatural head-to-body proportions, such as heads or facial features being disproportionately large, small, or misaligned with the body.
Visible blending or merging errors between the face and background, where facial features (such as hair, ears, or accessories) become difficult to distinguish from the surrounding areas or background.
Artifacts involving multiple faces or surreal features, where more than one head or body part appears in the image, indicating manipulation.
Distortion or loss of clarity in hair or accessories, such as overly stretched or distorted hair, or accessories that appear unnaturally merged with the face or head.
Unnatural facial expressions, such as awkward or stiff smiles or expressions that seem out of place.
Unnatural transitions between the head and neck, resulting in disjointed or distorted body proportions, or any misalignment between the head and neck.

If any of these severe artifacts are detected, respond directly with: severe artifacts.
If no severe artifacts are present and the image appears normal, respond with: normal.

Figure 9: Artifact Perception Prompts in Qwen2.5-VL.

## A.3 FAILURE CASES ANALYSIS

Although our method is generally stable and produces high-quality samples in the vast majority of cases, there are still a few failure cases under certain conditions. We identify two main types of failures:

| Matching Prompt for Standard Image | Matching Prompt for Non- Standard Image |
|---|---|
| Please determine if the two images depict the same person, focusing on the consistency of hairstyle, clothing, gender, and skin tone. Ensure the head pose and hair direction are naturally aligned, but minor variations in flow or positioning are acceptable. If the face is obscured, make a comprehensive judgment based on features such as hairstyle, accessories, or clothing. Differences caused by angle, lighting, or slight pose variations should be considered normal variations. Image A attributes: {attrs1} Image B attributes: {attrs2}

Output format: YES or NO, followed by one concise sentence explaining the key reason. | Determine if two images depict the same person: Minor differences in hairstyle, clothing, and accessories are allowed (such as slight changes in hairstyle volume or clothing wrinkles); gender and skin tone must be completely consistent; important accessories (glasses, hats, etc.) must not be added or removed; and there should be no obvious contradiction between head rotation and hairstyle direction. When the face is partially obscured, a comprehensive judgment can be made based on overall features. Image A attributes: {attrs1} Image B attributes: {attrs2}

Output format: YES or NO, followed by one concise sentence explaining the key reason. |

Figure 10: Matching Prompts in Qwen2.5-VL.

First, as shown in fig. 13 (a,b), we occasionally observe abnormal shoulder postures. This is primarily due to the fact that this work focuses on 3D full-head generation, and it is difficult to precisely control the shoulder pose when using Flux to generate multi-view images from frontal inputs via prompts. As a result, the generated multi-view images often exhibit significant variations in shoulder posture. However, during training, these images with different shoulder poses share the same semantic condition, which can lead the model to attempt to generate multiple conflicting shoulder configurations under the same condition. This results in unrealistic or distorted shoulder shapes in the output.

Second, some rare hairstyles or accessories (e.g., hats) are challenging to learn due to their high diversity and limited availability in the dataset. In such cases, the model may produce implausible outputs when generating under those semantic conditions, as illustrated by the unrealistic accessory in fig. 13 (c).

In addition, we observed extremely rare instances where the generated head is incomplete, as shown in fig. 13 (d). This may be attributed to segmentation errors in a small number of training images. When the model uses semantic conditions derived from such mis-segmented images, it tends to produce incomplete head structures.

Interestingly, despite these failures, the overall head shape remains realistic in most cases, indicating that our BalanceHead model has strong generalization capabilities. Even under outlier semantic conditions, the model is able to generate reasonable 3D head shapes with minor imperfections.

### A.4 SUPPLEMENTARY VIDEO

We provide rendered videos of 3D human heads generated by our model for the readers' reference. In total, there are 5 videos. `samples_group_1.mp4` to `samples_group_4.mp4` show 3D human heads randomly sampled from our BalanceHead model. We provide rendering videos covering 360-degree views, including changes in pitch angle. Each video contains either 12 or 24 samples.

In addition, we provide a video called `interpolations.mp4`, which visualizes interpolations between different randomly sampled 3D human heads from our model. Similarly, it includes rendering videos with 360-degree views and pitch angle variations.

For comparison, we include three videos with results from baselines: `baseline_hyplanedead_samples_random_view_condition.mp4`, `baseline_panohead_samples_random_view_condition.mp4`, and `baseline_spherehead_samples_random_view_condition.mp4`. These videos are generated using the official pretrained models under *random-view* conditioning, rather than the fixed front-view conditioning used in their original papers and demos. All results are produced with random seeds from 0 to 31, *without any cherry-picking*.

By comparing these videos, we make the following observations: First, the outputs from these three models are highly unstable. Some samples appear artifact-free, typically when a frontal view is randomly selected as the conditioning input; while others exhibit severe distortions or obvious artifacts, usually caused by non-frontal conditional views. Second, the diversity in identity, expression, hairstyle, and other attributes is limited. Although the appearances vary across samples, their underlying 3D geometry remains strikingly similar. Third, many distorted faces look nearly identical. This occurs because, in view-conditioned 3D-aware GANs, the generator's capacity and output diversity are strongly dependent on the conditioning view: diversity is high in the conditioned view but significantly reduced in non-conditioned views. Consequently, for samples conditioned on back views, the back sides may differ, but their front views often converge to similar appearances.

In contrast, our method is entirely free from this issue. By conditioning on a view-invariant semantic feature, we prevent any view-specific information from leaking into the generator during training. As a result, our model achieves consistent quality and diversity across all viewing directions, effectively resolving the view-dependency problem.

## A.5 More Random Samplings

fig. 15 shows additional randomly sampled full-head images from the latent space of BalanceHead. Each group contains six images, rendered at yaw angles of 0°, 45°, 90°, 135°, and 180°, with the last image displayed as a randomly sampled view.

## A.6 Discussion

### A.6.1 Impact of Dataset Size and Multi-View Images

Our method uses the CLIP feature of the front-view image as the semantic condition, so a corresponding frontal view is required for every training sample. Moreover, the multi-view collection ensures consistent data distributions across viewpoints, e.g., attributes such as ethnicity and hairstyle appear with similar frequencies in all views. This consistency helps the model generate 3D heads that are globally coherent. Therefore, using multi-view images is necessary for our method.

Regarding dataset scale, larger data primarily improves generation quality for rare cases. For example, with 1M training images, common head appearances are already well modeled, but rare styles (e.g., braids or hats) often lead to artifacts, because some specific hat or hairstyle variants appear in only dozens of training images. When scaling up to 11.2M images, the model sees significantly more examples of these rare cases, leading to markedly better and more robust generation.

### A.6.2 Non-frontal single-view GAN inversion

Our method is not limited to strictly frontal images. Specifically, we perform single-view GAN inversion using Pivotal Tuning Inversion (PTI) Roich et al. (2022), which optimizes both the latent code $w$ and generator parameters to match the target image via a combination of pixel-wise L2 loss and image-level LPIPS loss. Critically, this inversion process is independent of the semantic condition $c$, i.e., it does not involve predicting or optimizing $c$. Therefore, like prior methods, our approach can, in principle, handle target images from any viewpoint.

However, a practical limitation arises from head pose estimation: for large pose variations (e.g., profile views and back views), estimated poses are often inaccurate. And PTI relies on an accurate head pose to align the rendered output with the target image. For this reason, frontal views are typically used to evaluate inversion performance, to avoid confounding factors from pose estimation errors. We also include additional inversion results on non-frontal images in fig. 14 to demonstrate real-world applicability.

### A.6.3 Conditioning vs. Dataset Improvements

The core contribution of our work is the use of a view-invariant semantic condition, rather than conditioning on view as in prior work. This design is the key reason for our method's success, which is why we highlight it in the title, not the dataset. To validate the necessity of this semantic conditioning, we trained two ablation baselines on the same FLUX-generated dataset:

- Conditioning only on view (i.e., camera parameters), as in previous methods. This baseline aims to demonstrate that, even when trained on the new dataset, conventional view-based conditioning cannot resolve the directional bias and limited diversity issues.

- Conditioning on both view and semantic features. This baseline aims to show that removing view information from the conditioning is necessary to achieve view-invariant generation.

As shown in fig. 5(f,g), both baselines exhibit strong view-dependent bias: they produce realistic results at the conditioned view but suffer from noticeable distortions at other views. Table 1 quantifies this trend: both achieve low FID at the conditioned view (FID-view) but significantly higher FID on random views (FID-random). These results confirm that even with the improved dataset, the view-invariant semantic condition is essential for consistent, high-quality 3D-aware generation across all viewpoints.

### A.6.4 QUALITY OF FLUX-GENERATED IMAGES

We randomly visualize additional examples from the FLUX-generated dataset in fig. 12 of the revised version to better illustrate its quality and diversity. As shown, FLUX.1-Kontext generates non-frontal views that preserve key personal attributes from the frontal image, such as identity, expression, hairstyle, and lighting. As discussed in section 3.2, while these images are not strictly multi-view renderings with perfect 3D consistency, they are sufficient to be compatible with our training pipeline, which ultimately produces 3D full-heads with strong 3D consistency.

### A.6.5 ON THE NUMBER OF VIEWS AND DATASET DESIGN

Our method does not require a strictly multi-view dataset. The primary purpose of our data collection is to obtain front-view image CLIP features as the condition to train our view-invariant semantic-conditioned 3D-aware GAN. In fact, only a single paired sample per identity is minimally required, because any non-frontal view along with its corresponding frontal image is sufficient for training. In practice, however, we generate multiple views per frontal image for two key reasons:

- Computational efficiency: Generating a frontal image satisfying our selection criteria is significantly more expensive than generating non-frontal views. As described in section 3.1, we need to select the best frontal candidate from multiple generations, making one frontal image cost several times more than a single non-frontal view. To maximize data utility under limited compute, we generate as many non-frontal views as possible per valid frontal image, rather than just one.

- Distributional consistency: Using multiple views per identity ensures that the data distribution (e.g., hairstyle, ethnicity, expression) remains balanced across all viewpoints. For example, in an extreme case where short hair dominates the left views while long hair dominates the right views, the model may produce implausible "chimeric" heads, i.e. , short on one side and long on the other, which is also mentioned in SphereHead. Generating multiple views per identity mitigates this risk, as all views preserve the same identity-related attribute distribution as the frontal view.

Finally, controlling view generation with FLUX.1-Kontext is inherently noisy. Even when prompting for a specific view (e.g., "left view (yaw=90°)"), the model often outputs images at unintended yaw angles (e.g., 30°, 75°, or even 145°), and occasionally generates completely mismatched views (e.g., front, back or right when "left" was requested). Thus, even if we restrict prompts to fewer directions (front, back, left, right), the actual output still spans a continuous range of poses, making it impractical to enforce a fixed, small set of discrete views.

### A.6.6 CLARIFICATION ON VIEW SELECTION FOR SEMANTIC CONDITIONING

No single view captures 100% of the full-head semantic information. However, among all viewpoints, the frontal view provides the most comprehensive signal. It fully captures the facial region, which is most critical to human perception, and also conveys substantial global cues about hairstyle, hair color, clothing, and overall appearance. For example, if the frontal view shows a red afro hairstyle, the back view is highly likely to show the same red afro, but not a green ponytail. In other words, while the frontal view may miss some fine details (e.g., back-of-head hair geometry), it still

encodes the majority of identity-relevant semantics, whereas side or rear views capture significantly less, often lacking facial identity entirely. Therefore, we use the frontal view as the source of semantic conditioning, not because it is perfect, but because it is the most informative single view available.

### A.6.7 SEMANTIC CONDITIONING FROM A SINGLE FRONTAL VIEW

We do not fuse semantic conditions from multiple views not because of inference constraints (PTI optimizes only the latent code $w$, without predicting or optimizing the condition $c$), but for the following key reasons:

- Frontal view quality is significantly more reliable. Our data pipeline generates multiple frontal candidates and selects the one with the most accurate pose, highest visual quality, and strongest identity alignment with the original real image (see fig. 3). In contrast, non-frontal views suffer from unstable pose control (e.g., requesting a left profile sometimes yields yaw = 45° instead of 90°) and occasional identity drift or artifacts, despite post-generation filtering. Thus, non-frontal views are less trustworthy as semantic sources.

- Semantic consistency across views is anchored to the frontal image. All non-frontal views are generated from the selected frontal image using view prompts. This ensures minimal semantic drift between views. Using multiple independent views as conditions would introduce random inconsistencies (e.g., hairstyle, identity, expression mismatches), leading to conflicting semantic signals and unstable conditioning.

- Computational and architectural overhead. Our generator uses a StyleGAN2 backbone, where the condition $c$ (a 512-dimension CLIP feature) and the random number $z$ are projected into latent space $W$ via an MLP. Concatenating features from multiple views would drastically increase both parameter count and computational cost, with diminishing returns.

- The frontal view is already the most informative single view. As explained in our response to Section A6.6, while no single view captures 100% of full-head semantics, the frontal view contains the majority of identity-critical information, including face, hairstyle, color, and global appearance, which makes it the optimal choice for semantic conditioning.

For these reasons, we use only the frontal view to extract the semantic condition. Content in section 4.2 is also informative for this discussion.

### A.6.8 WHY IS THE SHARED FRONT-VIEW IMAGE CLIP FEATURE VIEW-INVARIANT

In our context, "align" and "anchor" refer to the process of using the frontal view corresponding to a given non-frontal image as the source for semantic conditioning. The logic is:

- We need a condition to facilitate training, or it will suffer from mode collapse, Fig. 2(a–c).

- However, traditional view conditioning leads to the directional bias issue Fig. 2(d–i).

- So, we need a view-invariant condition.

- It is inherently difficult to extract a truly view-invariant feature from a single image, as the content of any image inherently encodes its viewing perspective.

- We use (align/anchor) the random view image's corresponding frontal-view image to extract the feature as the semantic feature.

- Since all views derived from the same frontal image share the same semantic feature, this feature is de facto view-invariant.

- Experiments prove that using such a view-invariant semantic feature as the condition effectively prevents mode collapse and solves the directional bias issue.

Sections A.6.6 and A.6.7 further justify why the frontal view is the optimal choice for anchoring, due to its richness in identity-critical information and stability under our data generation pipeline.

### A.6.9 Generalizability to Other 3D-Aware Generators

All results presented in the paper, including ablation baselines, are based on HyPlaneHead. However, our proposed view-invariant semantic conditioning is compatible with earlier 3D-aware generator architectures, such as tri-plane (EG3D), tri-grid (PanoHead), and single or dual spherical tri-plane (SphereHead), as they all share the same StyleGAN2-based training pipeline, the only difference being their 3D representation.

We present qualitative and quantitative comparisons in fig. 16 and table 3. The results reveal consistent trends across different architectures: integrating our semantic conditioning effectively mitigates directional bias, similar to the behavior shown in fig. 2(d–i), and leads to significantly more diverse outputs.

As illustrated in fig. 16, all tri-plane-like representations benefit from our semantic conditioning, though they exhibit distinct characteristics. For tri-plane, when combined with our view-balanced dataset and semantic conditioning, the feature penetration issue is substantially alleviated. Nevertheless, some results still occasionally display excessive symmetry, as seen in (a). Moreover, due to the limited capacity of the tri-plane representation, outputs frequently suffer from blurriness (b,c) and structural adhesion, particularly around the chin area, as in (d).

For tri-grid, the severe mirroring artifacts are largely eliminated and no longer visually apparent. However, subtle feature penetration can still lead to unnatural hair textures, particularly in the back-of-head region opposite the jawline, as in (e). Furthermore, because tri-grid cannot be integrated with the unify-split strategy proposed in HyPlaneHead, inter-channel feature leakage introduces several characteristic artifacts. For instance, (f) exhibits crisscross, fishing-net-like patterns, while (g) shows horizontal ridges on the back of the head. These artifacts do not appear in our hy-plane-based main experiments. Additionally, some samples suffer from global incoherence. For example, in (h), the front of the hair appears purple while the back is black. We hypothesize that this stems from tri-grid's use of a larger number of feature planes, which makes maintaining cross-plane consistency more challenging. In contrast, hy-plane primarily encodes head information through a single spherical plane, thereby promoting better global coherence.

Interestingly, tri-grid also demonstrates a notable strength: it produces significantly more stable shoulder regions without the artifacts described in section A.3, thanks to its enhanced representational capacity in corner areas.

Finally, both single and dual spherical tri-plane representations tend to generate coarser hair strands and weaker fine-grained details, as shown in (k-n). (i,j) In the single spherical variant, although seam artifacts become less pronounced, the representation capacity near the seam is constrained, often resulting in messy or distorted hair in the back-of-head region. (o,p) The dual spherical version, while more expressive, still occasionally exhibits disordered hair with horizontal and vertical striping, suggesting residual inter-channel feature penetration.

Besides, though at a lower frequency, multiple-face artifacts persist across all representations, including HyPlane. We further validate that incorporating the ViCiCo loss quickly resolves the multiple-face artifacts (typically within a few hundred kimg), consistent with prior work. In summary, our semantic conditioning generalizes well to various 3D generator backbones, consistently improving output quality and diversity. Our main experiments use the most advanced representation (HyPlaneHead), which achieves the best overall performance.

### A.6.10 Baselines Using Identity Features for Conditioning

We introduce two additional identity-conditioned baselines to analyze the role of semantic conditioning in our framework.

- Conditioning on the identity feature extracted from the current random-view image. Results: FID-random: 16.37, FID-front: 4.88.
- Conditioning on the identity feature extracted from the corresponding front-view image of the current random view. Results: FID-random: 5.06, FID-front: 4.38.

In the first baseline, the identity (ID) feature is less view-variant than the CLIP image feature but is not fully view-invariant. Specifically, for the same person, the ID feature remains relatively sta-

Table 2: Comparison with state-of-the-art methods.

| Method | PanoHead | SphereHead | HyPlaneHead | **Ours** |
|---|---|---|---|---|
| **MKNND** $\uparrow$ | 0.391 | 0.345 | 0.291 | **0.456** |
| **LPIPS** $\downarrow$ | 0.086 | 0.083 | 0.084 | **0.076** |

ble under near-frontal views; however, it changes dramatically under large-profile or back views. Moreover, ID feature extraction is highly sensitive to image preprocessing, such as cropping and rotation, and typically relies on a head detector and alignment model for stable estimation. Unfortunately, these models are not robust to extreme poses and often fail entirely on back views. To enable training on such views, we must forgo alignment as a compromise: when head detection fails, we directly feed the raw image into the ArcFace ID feature extractor. This makes the ID feature highly unstable across views. Since the ID feature implicitly encodes both semantic identity and residual view information, it behaves similarly to the "view-semantic" conditioning in Table 1, resulting in a high FID-random. However, because the ID feature is stable in near-frontal views, it functions comparably to the frontal view CLIP feature in those cases, yielding a low FID-front.

In the second baseline, all non-front views of the same subject share the same front-view-derived identity feature. This shared feature is effectively view-invariant and plays a role nearly identical to the shared CLIP semantic feature used in our main experiments, leading to low FID scores on both random and front views. However, since identity features primarily capture facial characteristics and pay little attention to accessories such as clothing, hats, or hairstyle, models conditioned on them tend to generate blurry or incoherent reconstructions in these regions.

### A.6.11 DIVERSITY COMPARISON

We compare our method against state-of-the-art approaches in terms of diversity. To mitigate the influence of outliers and ensure a robust evaluation, we adopt the *Mean k-Nearest Neighbor Distance* (MKNND) with $k = 5$ in the ArcFace embedding space as our diversity metric. Specifically, let $\{\mathbf{z}_i\}_{i=1}^N$ denote the ArcFace identity features of $N$ generated samples. For each sample $\mathbf{z}_i$, we compute its cosine distance to all other generated samples and identify its $k$ nearest neighbors $\{\mathbf{z}_j\}_{j \in \mathcal{N}_k(i)}$, where $\mathcal{N}_k(i)$ is the set of indices of the $k$ closest samples to $\mathbf{z}_i$ (excluding itself). The MKNND is then defined as:

$$\text{MKNND} = \frac{1}{N} \sum_{i=1}^N \left( \frac{1}{k} \sum_{j \in \mathcal{N}_k(i)} d_{\cos}(\mathbf{z}_i, \mathbf{z}_j) \right), \tag{2}$$

where $d_{\cos}(\mathbf{a}, \mathbf{b}) = 1 - \frac{\mathbf{a}^\top \mathbf{b}}{\|\mathbf{a}\|\|\mathbf{b}\|}$ is the cosine distance between two feature vectors. A higher MKNND indicates that the generated identities are more dispersed and less clustered in the embedding space, reflecting greater identity diversity. As shown in table 2, our method achieves state-of-the-art performance in diversity compared to existing methods.

### A.6.12 QUANTITATIVE GAN INVERSION COMPARISON

To quantitatively compare reconstruction fidelity with existing methods, we perform single-view GAN inversion on a test set of 100 in-the-wild images. We then compute the LPIPS distance using features extracted from the target image and the rendered image from the same viewpoint. Since methods like PTI optimize model parameters for detailed per-image fitting, as shown in table 2 and fig. 6, all approaches achieve visually realistic reconstructions at the input viewpoint. Although the LPIPS scores are close, our method achieves the lowest value, demonstrating its superior reconstruction fidelity and practicality in real-world scenarios.

### A.6.13 NEAREST-NEIGHBOR ANALYSIS

As shown in fig. 17, we perform a nearest-neighbor analysis by visualizing, for each generated sample, the most similar real image from the training set. Similarity is measured using the cosine distance between the CLIP feature of the frontal view of the generated 3D head and the CLIP features of real training images. In every row, the left side is the generated image, and the right side is its nearest neighbor in the training set.

A.6.14   FUTURE WORK

In this paper, we have presented a new state-of-the-art full-head generative model, demonstrating the significant potential of inconsistency-tolerant 3D training strategies and robust semantic-conditioned models. By effectively leveraging data that lacks strict multi-view consistency, our approach remarkably enhances both generative quality and diversity. Looking ahead, we aim to extend these strategies beyond 3D-aware GANs to broader architectures, such as Large Reconstruction Models (LRM) Hong et al. (2023); Tang et al. (2024); Qiu et al. (2025a;b) and diffusion-based models Wang et al. (2023); Liu et al. (2023). Furthermore, as a foundational 3D full-head model, BalanceHead provides a smooth and well-structured latent space that facilitates various downstream applications, including portrait animation and talking head synthesis Ma et al. (2023); Sun et al. (2023); Tan et al. (2024b); Deng et al. (2024); Wu et al. (2023b); Tan et al. (2024a; 2025), stylization Zhang et al. (2023); Jiang et al. (2023), and editing Sun et al. (2022); Zhang et al. (2024); Haque et al. (2023). These directions will be the focus of our future research.

Table 3: Compatibility of semantic conditioning with different tri-plane-like representations

| Representation | tri-plane | tri-grid | single sph-tri-plane | dual sph-tri-plane | hy-plane |
|---|---|---|---|---|---|
| **FID-random** | 5.46 | 4.42 | 4.85 | 5.27 | **3.67** |

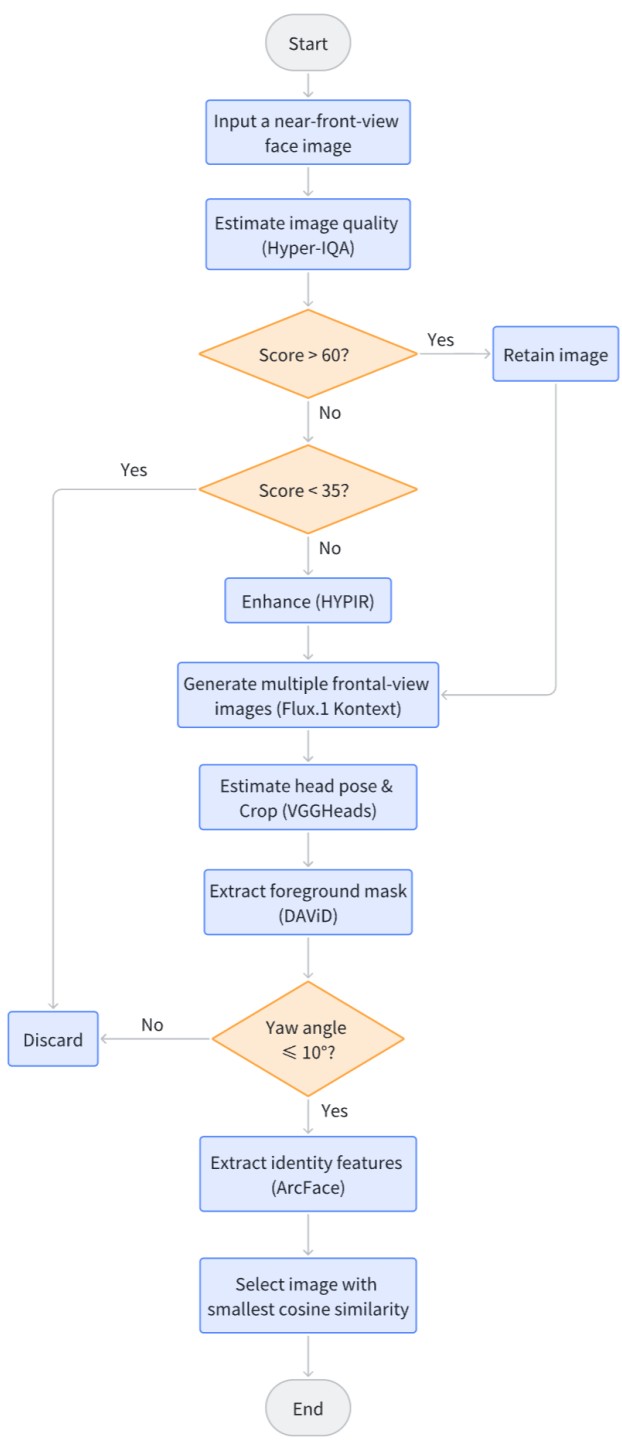

Figure 11: The Front-view Image Generation and Preprocessing Pipeline.

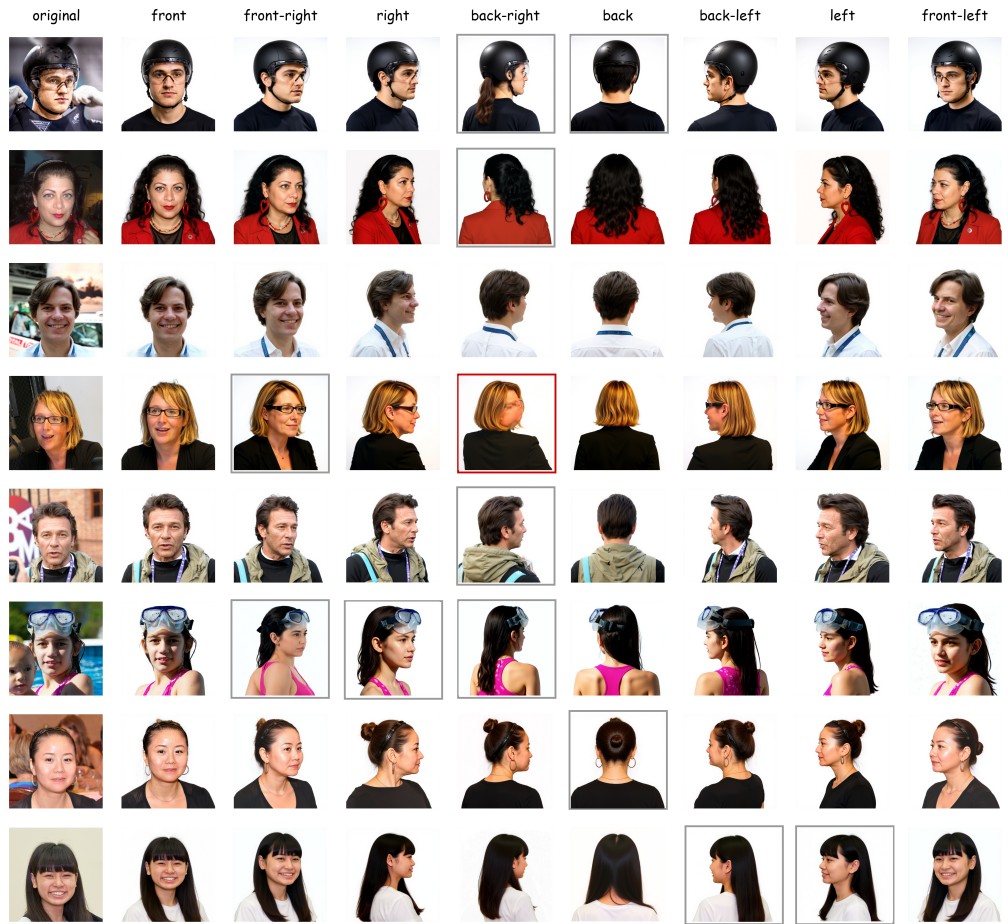

Figure 12: Samples from our training dataset generated by FLUX.1-Kontext. Images enclosed in gray bounding boxes exhibit global incoherence with their corresponding front-view references, while those in red bounding boxes contain severe visual artifacts.

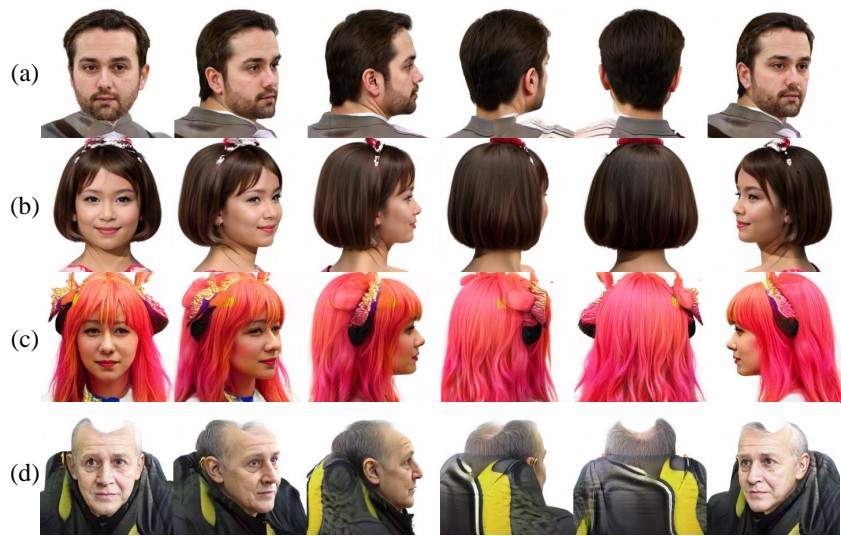

Figure 13: Failure Cases.

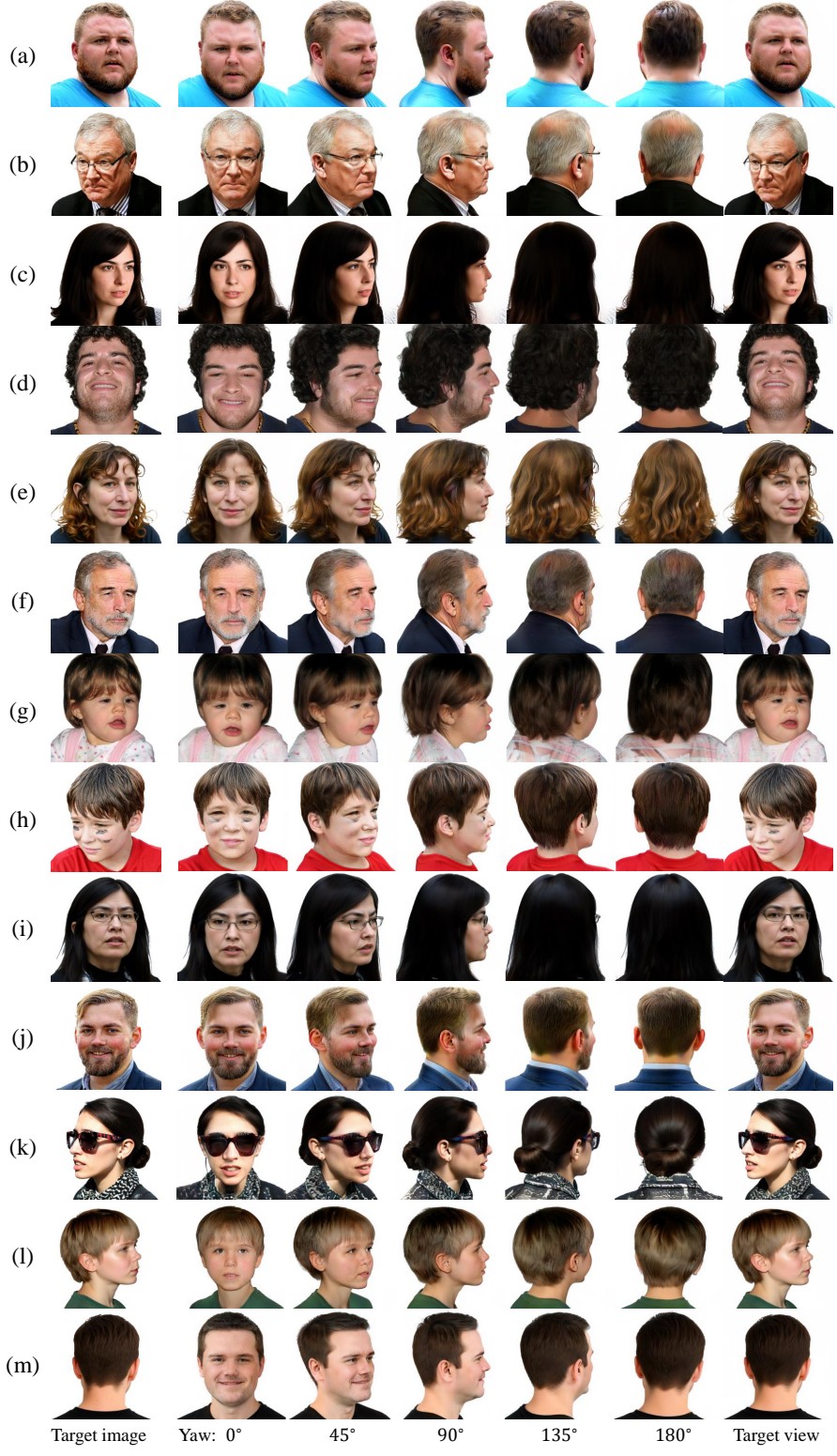

Figure 14: Single-view GAN inversion using Pivotal Tuning Inversion (PTI). (a–j) Inversion results for near-frontal target images. (k–m) Inversion results for large-pose target images.

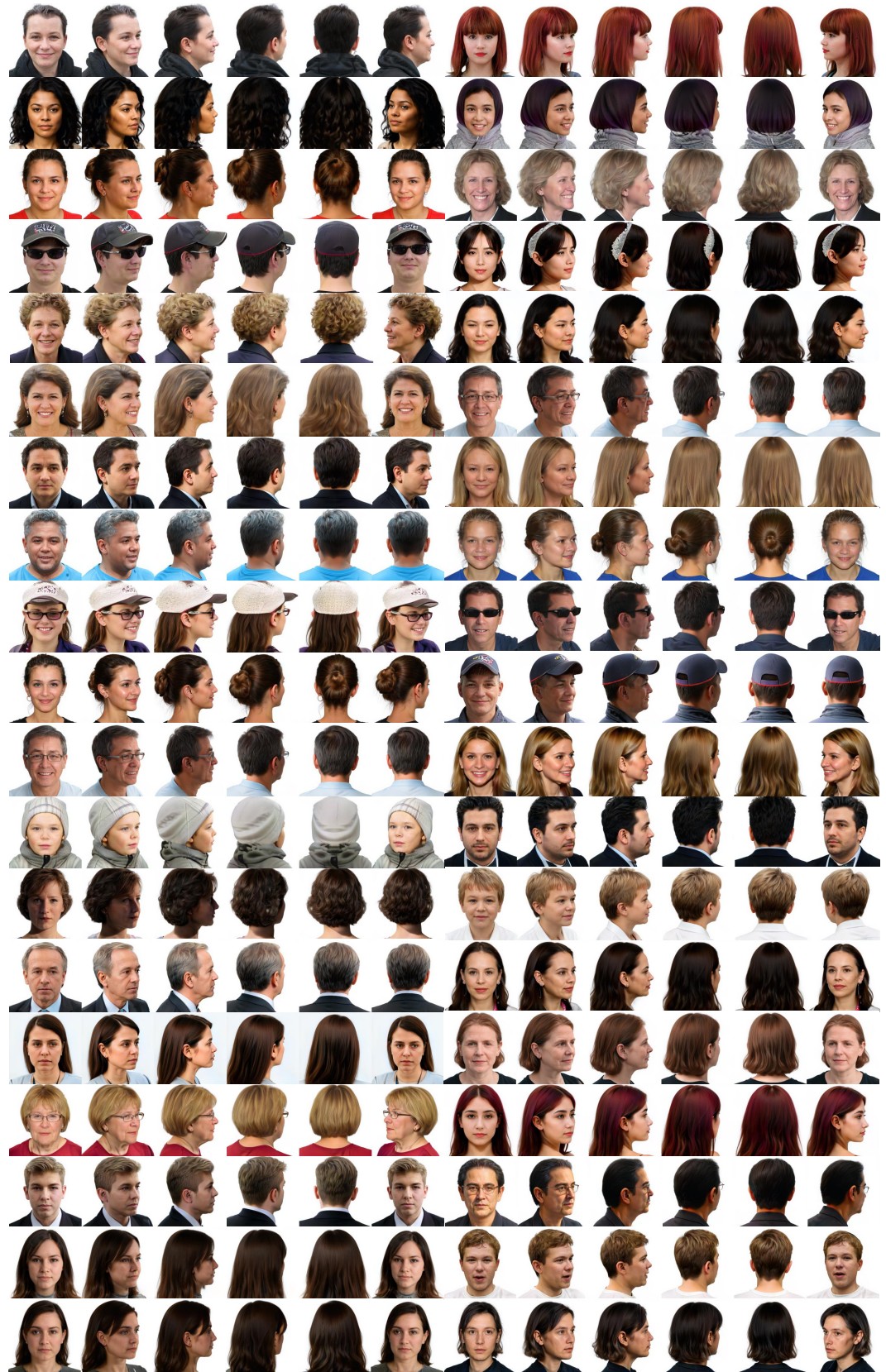

Figure 15: Random Sampling Results.

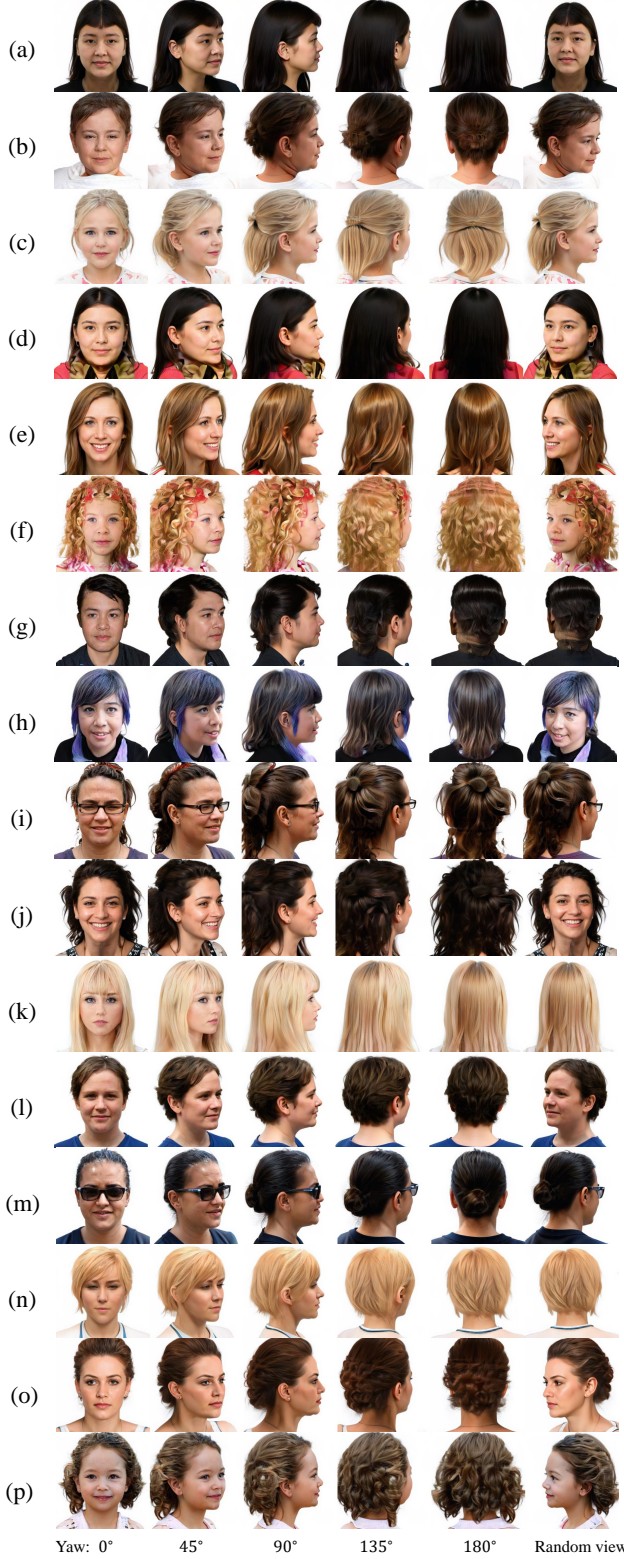

Yaw: 0°        45°        90°        135°        180°        Random view

Figure 16: Compatibility with other tri-plane-like representations. (a-d) Tri-plane representation. (e-h) Tri-grid representation. (i-l) Single spherical tri-plane representation. (m-p) Dual spherical tri-plane representation.

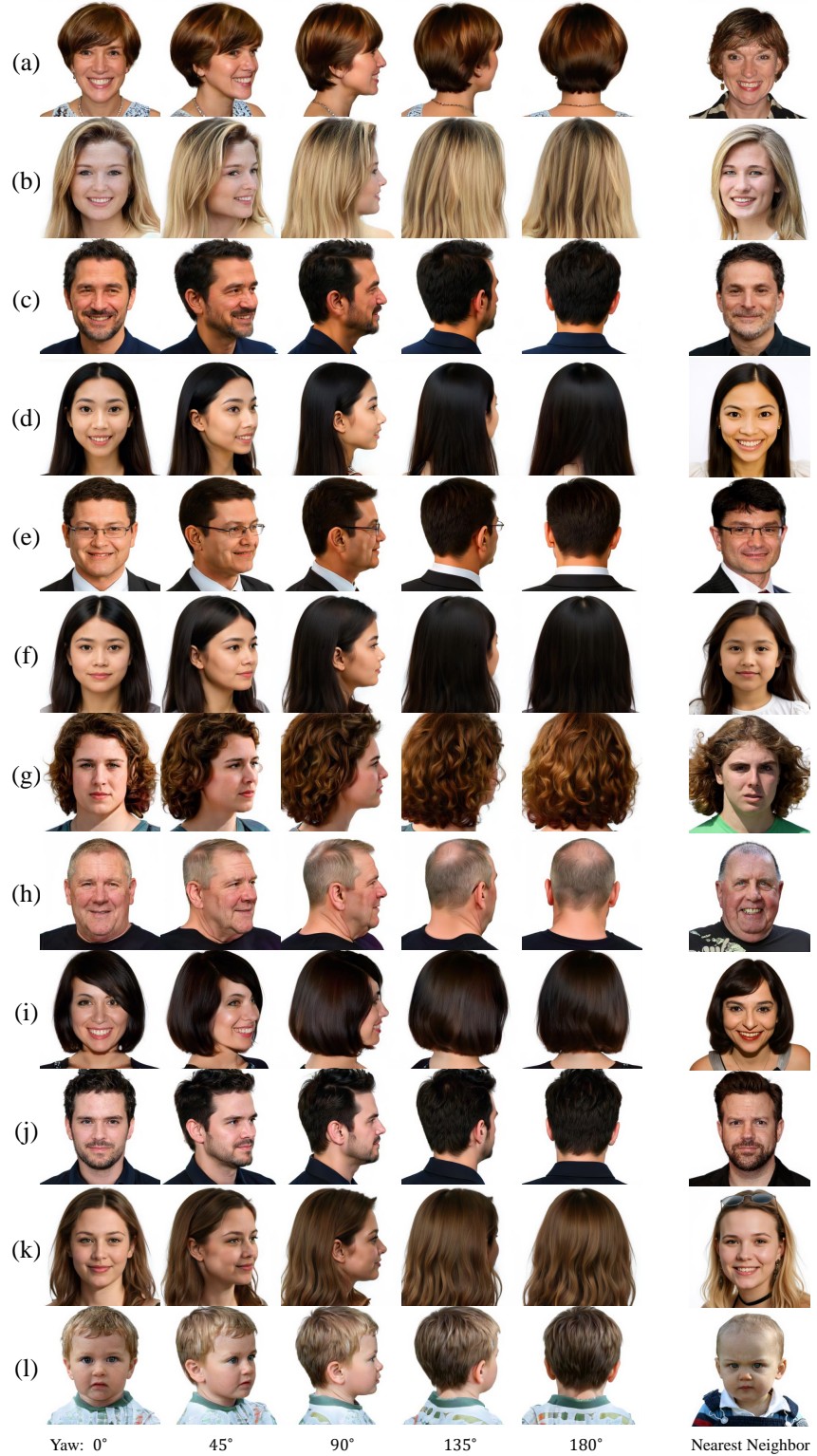

Yaw: 0°        45°        90°        135°        180°        Nearest Neighbor

Figure 17: Nearest-neighbor analysis. The left side is the generated image, and the right side is its nearest neighbor in the training set.

