# OpenReview forum: "Condition Matters in Full-head 3D GANs"
_ICLR.cc/2026/Conference — ICLR 2026 Poster_

### Official Review · Reviewer_VqNs · 2025-10-25

**Soundness:** 3
**Presentation:** 2
**Contribution:** 2
**Rating:** 6
**Confidence:** 5

**Summary:**

The authors first create a dataset with a range of view angles using FLUX.1 Kontext - 10 multi-view head images. Therefore, they start with existing front-view datasets first, and then use FLUX.1 Kontext to predict the respective other views. While the 2D prior is not 3D consistent but it preserves the identity which is sufficient for training their 3D-aware GAN.

The paper then proposes a method similar to 3D-aware GAN head generator, in the spirit of existing works such as PI-GAN, EG3D, SphereHead, PanoHead, GGHead, HyPlane etc. The latter approach is the closest method. Here, the authors also propose a semantic-conditioned 3D-GAN head generator. The front view is used to anchor all image vies as a single reference image. Semantic features are then used as a condition – in their current versions the authors use CLIP as a feature embedding.

Overall, the method is essentially HyPlane with a semantic conditioning, trained on a new MVS FLUX-generated multi-view dataset.

**Strengths:**

- Nice results
- The dataset is quite valuable (is it being released?)

**Weaknesses:**

- technically quite incremental over previous methods such as SphereHead or HyPlane
- poor evaluation (see below)

**Questions:**

Comparison are somewhat unfair as GGHead, PanoHea, and SphereHead are trained on different data.
- What if the proposed method was trained on the same data as these baselines?
- What if the baselines were trained on the proposed dataset?

I would’ve also loved to see the comparisons in the video. At the moment, the only comparisons are shown in Figure 5, which is insufficient to properly evaluate the method.

Somewhat optional but a nearest neighbor / novelty analysis with respect to the train images would be nice. I would expect this to work well though.

The writing in the quantitative results paragraph (end of 5.1) is a bit confusing; it would be great if it could be checked for clarity. For instance, what features are used for the FID evaluation? Are they used from the generated dataset? Could you please clarify.

---

> ### Author Response · Authors · 2025-11-27
>
> # W1: Incremental technical contribution
> We emphasize that our core contribution is the view-invariant semantic conditioning, not the dataset itself. The proposed semantic conditioning is non-trivial and essential: simply replacing traditional view-based conditioning (e.g., camera parameters) with our semantic condition is what enables high-fidelity, diverse, and consistent 3D-aware generation across all views.
>
> Our detailed analysis reveals that the choice of conditioning plays a critical yet previously overlooked role in the training stability and output diversity of 3D-aware GANs; remarkably, this long-standing challenge can be effectively addressed by the simple yet principled shift from view conditioning to view-invariant semantic conditioning.
> Without this innovation, even when trained on our new dataset, models still suffer from directional bias and limited diversity, as shown in our ablation studies.
>
> The purpose of the new dataset is to support training of our semantic-conditioned GAN: it provides, for every non-frontal image, a corresponding high-quality frontal view to extract a view-invariant semantic feature. This pairing is necessary to implement our conditioning strategy but is not the main technical advance.
>
> To validate the necessity of this semantic conditioning, we trained two ablation baselines on the same FLUX-generated dataset:
> 1. Conditioning only on view (camera parameters), as in previous methods
> 2. Conditioning on both view and semantic features.
> As shown in Fig. 5(f,g), both baselines exhibit strong view-dependent bias: they produce realistic results at the conditioned view but suffer from noticeable distortions at other views. Table 1 quantifies this trend: both achieve low FID at the conditioned view (FID-view) but significantly higher FID on random views (FID-random).
> These results confirm that even with the improved dataset, the view-invariant semantic condition is essential for consistent, high-quality 3D-aware generation across all viewpoints.
>
>
> # Q1: Comparison Fairness
> Our method relies on extracting a view-invariant semantic feature from the corresponding frontal-view image of each training sample. This is not feasible with the datasets used by prior works (e.g., GGHead, PanoHead, SphereHead), which typically consist of single-view head images collected from the internet, making it impossible to obtain the required frontal counterpart for arbitrary input views.
>
> To ensure a fair comparison, we retrain several baseline methods (PanoHead, SphereHead, and HyPlaneHead) on our proposed dataset. Note that HyPlaneHead trained on our data corresponds exactly to the first row in Table 1. These results are reported in Supp. A.6.9.
>
> Please note that GGHead’s UV mapping does not cover the back of the head. Consequently, even when trained on our dataset, it cannot represent the rear head region and cannot function as a full-head model. For this reason, we exclude it from our comparisons.
>
> Crucially, even when trained on our high-quality multi-view dataset, these baselines still suffer from directional bias and limited diversity, unless equipped with our view-invariant semantic conditioning. This confirms that the performance gain stems primarily from our conditioning design, not just the dataset.

---

> ### Author Response · Authors · 2025-11-27
>
> # Q2: More video comparison
> For comparison, we include three videos with results from baselines:
> baseline_hyplanedead_samples_random_view_condition.mp4,
> baseline_panohead_samples_random_view_condition.mp4, and
> baseline_spherehead_samples_random_view_condition.mp4.
> These videos are generated using the official pretrained models under random-view conditioning, rather than the fixed front-view conditioning used in their original papers and demos. All results are produced with random seeds from 0 to 31, *without any cherry-picking*.
>
> By comparing these videos, we make the following observations:
>
> 1. the outputs from these three models are highly unstable. Some samples appear artifact-free, typically when a frontal view is randomly selected as the conditioning input; while others exhibit severe distortions or obvious artifacts, usually caused by non-frontal conditional views.
> 2. the diversity in identity, expression, hairstyle, and other attributes is limited. Although the appearances vary across samples, their underlying 3D geometry remains strikingly similar.
>
> 3. many distorted faces look nearly identical. This occurs because, in view-conditioned 3D-aware GANs, the generator’s capacity and output diversity are strongly dependent on the conditioning view: diversity is high in the conditioned view but significantly reduced in non-conditioned views. Consequently, for samples conditioned on back views, the back sides may differ, but their front views often converge to similar appearances.
>
> In contrast, our method is entirely free from this issue. By conditioning on a view-invariant semantic feature, we prevent any view-specific information from leaking into the generator during training. As a result, our model achieves consistent quality and diversity across all viewing directions, effectively resolving the view-dependency problem.
>
> # Q3: nearest neighbor / novelty analysis
> Please refer to Supp. A.6.11 and Figure 17 for details.
>
> # Q4: Clarification of FID Evaluation Protocol
> Following prior works (e.g., EG3D, PanoHead, SphereHead, HyPlaneHead), we use the Inception-v3 network (https://api.ngc.nvidia.com/v2/models/nvidia/research/stylegan3/versions/1/files/metrics/inception-2015-12-05.pkl) to extract features for FID computation. Specifically, we randomly sample 50k images from the FLUX-generated dataset as the real reference set and generate 50k synthetic images using our method; the FID is then computed between these two sets.

---

### Official Review · Reviewer_9wYw · 2025-10-30

**Soundness:** 3
**Presentation:** 3
**Contribution:** 3
**Rating:** 6
**Confidence:** 4

**Summary:**

The paper introduces a novel approach for generating high-fidelity and diverse 3D human heads using Generative Adversarial Networks (GANs). The core contribution is the model, BalanceHead, which addresses the severe mode collapse that typically plagues unconditioned full-head 3D-aware GANs. The authors propose a novel conditioning strategy by integrating view-invariant semantic features into the model's training process. This conditioning is claimed to stabilize training and enable the generation of diverse, multi-view renderings and their corresponding 3D geometries

**Strengths:**

1. The work targets a well-known instability issue in 3D-aware GANs, specifically the severe mode collapse that results from a lack of proper conditioning.
2. The idea of using view-invariant semantic features is a promising direction for decoupling identity/semantics from viewing pose, which should inherently lead to more stable and diverse generation compared to simpler conditioning methods.
3. The  BalanceHead is shown to generate high-quality results, including random-view, multi-view renderings, and visualizations of the underlying 3D geometries, indicating successful 3D structure synthesis and view consistency.

**Weaknesses:**

1. Experiments do not offer a clear comparison to established State-of-the-Art full-head 3D-aware GANs. Without standard quantitative metrics (e.g., FID, diversity scores, LPIPS), the claimed high-fidelity and diversity are difficult to objectively verify.
2. While the conditioning is claimed to be novel, the paper needs to demonstrate the necessity of using view-invariant semantic features specifically. Without an ablation comparing this approach to simpler, non-semantic conditioning or existing view-variant conditioning methods, the complexity of the proposed feature extraction is not fully justified.

**Questions:**

1. Please include a comprehensive quantitative comparison against relevant SOTA full-head 3D-aware GAN models. This should include metrics such as FID (Fréchet Inception Distance), Diversity/Recall scores, and potentially LPIPS or KID across multiple common head datasets.
2. Please compare the proposed view-invariant semantic features against simpler conditioning methods (e.g., using explicit pose/camera parameters alone, or using a simple identity embedding). And please  demonstrate the difference in performance (especially diversity/mode collapse) when using view-invariant versus view-variant features.

---

> ### Author Response · Authors · 2025-11-27
>
> Thank you for your valuable feedback. Your feedback will undoubtedly help improve our paper.
>
> # W1 & Q1: More Quantitative Evaluation
> Computing fair generative metrics, especially FID-related metrics, requires all methods to be trained on the same dataset, so we cannot directly use publicly released pretrained models from prior work.
> We evaluate our method and baselines using the following standard metrics:
> - Standard FID (FID-view): The standard FID score reported in prior works (e.g., EG3D, PanoHead, SphereHead, HyPlaneHead) evaluates generation quality only at the conditioned view. This is exactly the metric we report as “FID-view” in Table 1. We use this naming to distinguish it from our additional metric, FID-random.
> - FID-random: To assess full 360° synthesis quality and diversity, we follow HyPlaneHead and render images from random novel views. This metric is most relevant to our task, as FID-view ignores non-conditioned views. As shown in Table 1, our method achieves a significantly lower FID compared to the baselines, not only on random views, but also on the views that the baselines condition on.
> - Diversity: To evaluate the diversity (primarily identity diversity, as this is a head-related task) in generated images, we compute the Mean k-Nearest Neighbor Distance (MKNND) in the ArcFace embedding space. Specifically, for each generated sample, we measure its cosine distance to its k-nearest neighbors among all generated samples and report the average across the entire set. A higher MKNND indicates that the generated identities are more dispersed and less clustered, reflecting greater diversity. Results in Table 2 indicate that our method achieves state-of-the-art diversity compared to existing methods.
> - LPIPS: We use LPIPS to evaluate the reconstruction fidelity of the 3D head. Since our multiview training data is not strictly 3D-consistent, computing LPIPS on novel views from the training set would be meaningless. Instead, we apply LPIPS to assess the quality of GAN inversion by measuring the perceptual distance between the input single image and the rendered synthetic image from the same viewpoint. As shown in Table 2, the LPIPS results demonstrate that our method outperforms existing approaches in terms of reconstruction fidelity.

---

> ### Author Response · Authors · 2025-11-27
>
> # W2 & Q2: Ablation on Conditioning Design
> We agree that validating the necessity of our view-invariant semantic conditioning is essential. In the paper, we already include two baselines trained on the same data:
> - Conditioning only on view (i.e., camera parameters), as in previous methods. This baseline aims to demonstrate that, even when trained on the new dataset, conventional view-based conditioning cannot resolve the directional bias and limited diversity issues.
> - Conditioning on both view and semantic features. This baseline aims to show that removing view information from the conditioning is necessary to achieve view-invariant generation.
> As shown in Fig. 5(f,g), both baselines exhibit strong view-dependent bias: they produce realistic results at the conditioned view but suffer from noticeable distortions at other views. Table 1 quantifies this trend: both achieve low FID at the conditioned view (FID-view) but significantly higher FID on random views (FID-random).
> These results confirm that even with the improved dataset, the view-invariant semantic condition is essential for consistent, high-quality 3D-aware generation across all viewpoints.
>
> Following your suggestion, we add two additional baselines:
>
> 1. Conditioning on the identity feature extracted from the current random-view image. Results: FID-random: 16.37, FID-front: 4.88.
> 2. Conditioning on the identity feature extracted from the corresponding front-view image of the current random view. Results: FID-random: 5.06, FID-front: 4.38.
> In the first baseline, the identity (ID) feature is less view-variant than the CLIP image feature but is not fully view-invariant. Specifically, for the same person, the ID feature remains relatively stable under near-frontal views; however, it changes dramatically under large-profile or back views. Moreover, ID feature extraction is highly sensitive to image preprocessing, such as cropping and rotation, and typically relies on a head detector and alignment model for stable estimation. Unfortunately, these models are not robust to extreme poses and often fail entirely on back views. To enable training on such views, we must forgo alignment as a compromise: when head detection fails, we directly feed the raw image into the ArcFace ID feature extractor. This makes the ID feature highly unstable across views. Since the ID feature implicitly encodes both semantic identity and residual view information, it behaves similarly to the “view-semantic” conditioning in Table 1, resulting in a high FID-random. However, because the ID feature is stable in near-frontal views, it functions comparably to the frontal view CLIP feature in those cases, yielding a low FID-front.
>
> In the second baseline, all non-front views of the same subject share the same front-view-derived identity feature. This shared feature is effectively view-invariant and plays a role nearly identical to the shared CLIP semantic feature used in our main experiments, leading to low FID scores on both random and front views. However, since identity features primarily capture facial characteristics and pay little attention to accessories such as clothing, hats, or hairstyle, models conditioned on them tend to generate blurry or incoherent reconstructions in these regions.

---

### Official Review · Reviewer_t6Lb · 2025-10-31

**Soundness:** 3
**Presentation:** 3
**Contribution:** 3
**Rating:** 8
**Confidence:** 4

**Summary:**

This paper introduces a new conditioning method for full-head 3D GANs. Unlike previous work which uses camera view directions as the condition, this paper proposes to use semantic image features extracted from the frontal view images as the condition. To do so, the authors curated a large-scale synthetic portrait image dataset with balanced coverage of the view directions and identities, in which each identity has multi-view images generated with Flux. By training on this curated dataset with the proposed conditioning method, experiments report improved performance compared to existing 3D GAN methods.

**Strengths:**

[Originality]
- How to improve the quality of 3D full head synthesis has been a long problem since the publication of EG3D, and the inherited view conditioning has been a quite annoying part. This paper proposes a new conditioning method under the help of current 2D image generation and editing models. Though the data are synthetic, they maintain a level of realism and help to learn a more geometrically consistent 3D GAN. I personally like this idea.

[Quality]
- This paper conducts common qualitative experiments regarding 3D GAN, including random sampling, novel view synthesis, GAN inversion, and interpolation. The results look plausible to me, especially regarding the hair details.

[Clarity]
- The manuscript is easy to follow and understand. If you have basic knowledge of 3D GANs such as EG3D, you should be able to follow the details and solutions proposed in this paper.

**Weaknesses:**

Overall I do not see significant weaknesses of the current manuscript.

**Questions:**

I do not have major questions regarding the manuscript, only a small question:
1. Will the curated dataset be released? As far as I know, data curation should be the most time consuming and dirty part of this work. I’m glad to see the authors put lots of effort into solving it, and I believe the release of this dataset will facilitate further research in this community.

**Details Of Ethics Concerns:**

As this work involves portrait generation of humans, it should be really careful to ensure no harmful content is generated with this method.

---

> ### Author Response · Authors · 2025-11-27
>
> Thank you for your recognition of our paper.
>
> The dataset, together with the code for data generation and preprocessing, will be made publicly available upon acceptance of this paper.

---

### Official Review · Reviewer_WZgK · 2025-11-01

**Soundness:** 3
**Presentation:** 3
**Contribution:** 3
**Rating:** 4
**Confidence:** 5

**Summary:**

The method produces a training procedure for better training of GANs for generating 3D human heads, which are always 3D-consistent and have a realistic appearance from all sides, including the back of the head, and do not feature artefacts such as Janus artefacts or symmetry breaking at the back of the head. The method is based on an observation that view conditioning is suboptimal for training such GANs, as it introduces instability and 3D-inconsistency. The authors propose semantic conditioning based on a CLIP feature of an image from a frontal view. The results are demonstrated as unconditional sampling visuals and FID-* metrics. The method requires multi-view data to be trained. To obtain such, the authors propose to use FLUX diffusion model to generate such multi-view data and a curation pipeline to filter out unrealistic samples.

**Strengths:**

- Paper story sounds generally reasonable. Very clear pitch right from the abstract.
- The paper is interesting and, despite presenting a simple observation and a fix for it, has value for the community and provides an additional important insight into training GANs. For me, reading the paper was interesting mainly because of the observation about the GAN instability and a bit less about the proposed fix for it (see Weaknesses).
- Image-flitering head is a smart idea that ensures high quality of the results.
- Great unconditional sampling results, even for the failure cases in Fig. 12. Generally, the improvement in both metrics and visuals is very substantial.

**Weaknesses:**

- Not very clear how much the size of the dataset + presence of multi-view images in the dataset is actually helping.
- Writing is often unclear.
    - Teaser is also a bit unclear: the results there look just very similar to other generators, e.g., SphereHead. I think it would be clearer for the reader if some conditioning is also shown. Also, the geometry of the back of the head is not shown there, even though it's likely one of the strongest improvement points.
    - My (subjective) feeling from the text is that LLMs might have rephrased it (as authors fairly acknowledge), but in a way that is sometimes hard to understand and with repeated claims/sentences (see Questions). Some things like absent spaces around an em-dash ("—") -- e.g. see L313) clearly reveal the lack of post-editing after the use of LLMs.
- The fact that the frontal view feature is used means that the single-view GAN inversion can only be done for ideally frontal images, right? What happens if the head is at least slightly rotated? And when it's e.g. a profile view?
    - How exactly was GAN inversion done? Was it latent code optimization, PTI, or anything else?
- A bit raw experimental section (see Questions)

**Questions:**

- How much do we benefit from the condition itself, not from switching to a new FLUX-generated dataset?
    - If multi-view data is necessary for training of this method, one could do an ablation with respect to the size of the dataset and the number of views in it
- How good are the images generated by FLUX? Probably a visual would help to understand it (except one example in Figure 3)
- Even though the pitch is pretty clear, not sure why we can't apply it to the existing generator; do we also absolutely necessarily need an (orthogonal) contribution with FLUX data?
    - There's an explanation for that in the Intro: "Specifically, for a person’s multi-view images, we align the conditioning of all views to the semantic feature of the frontal view, since it contains the most comprehensive information" --> multi-view data is rare --> we generate it.  Two questions regarding that:
        - What does "we align" mean? (Or "we anchor" later in the text). This remains unclear after reading the whole manuscript too.
        - Does this sentence mean that the method can only work when there are multi-view images of a person available during inference (to obtain a frontal image feature to condition the GAN)? Or is it a requirement only in training? If only in training, then how is inference performed when the image is not exactly frontal? Perhaps it would make sense to train another network that would try to predict the frontal-view CLIP feature from other views that are close to the frontal? And then no multi-view data would be required in training.
- In any case, GAN inversion results of not exactly frontal images should be shown. Otherwise, the direct applicability of the method would be highly limited.
- Do I understand correctly that all the experiments are done for HyPlaneHead only? What happens if the trick is applied to other generators? For instance, does the semantic conditioning fix the PanoHead's issue of symmetry breaking in the back of the head, or is SphereHead's ViCiCo loss still crucial to achieve that?
- L269: "Nvidia A10 GPU" -- just to confirm: was A10 or A100 meant?
- Will the dataset be released publicly?
- L315: "frontal views capture rich facial features but lack hair details, while rear views emphasize hair but miss facial features." And then, L320: "Among all possible views, the front view contains the most comprehensive global information, because it includes not only the facial region, which is most sensitive to human perception, but also reflects overall appearance, clothing, and hairstyle." Isn't it illogical? Basically, first, the authors are saying that the back view is essential, and then they are saying that the frontal view contains enough info. I understand that the back view does not contain info (it's evident from the visuals), but the writing seems a bit misleading.
- Why not stack the semantic info from several (predefined) views then? Is it because we don't have those in inference?

---

> ### Author Response · Authors · 2025-11-27
>
> Thank you for your detailed review and valuable suggestions. Your feedback will undoubtedly help improve our paper. In addition to the responses below, we have added a new section, “A.6 Discussion,” in the supplementary material to address several of your questions in more detail.
>
> # W1: Impact of Dataset Size and Multi-View Images
> Our method uses the CLIP feature of the front-view image as the semantic condition, so a corresponding frontal view is required for every training sample. Moreover, the multi-view collection ensures consistent data distributions across viewpoints, e.g., attributes such as ethnicity and hairstyle appear with similar frequencies in all views. This consistency helps the model generate 3D heads that are globally coherent. Therefore, using multi-view images is necessary for our method.
>
> Regarding dataset scale, larger data primarily improves generation quality for rare cases. For example, with 1M training images, common head appearances are already well modeled, but rare styles (e.g., braids or hats) often lead to artifacts, because some specific hat or hairstyle variants appear in only dozens of training images. When scaling up to 11.2M images, the model sees significantly more examples of these rare cases, leading to markedly better and more robust generation.
>
>
> # W2.1 Unimpressive Teaser
> While our teaser results may appear visually similar to prior works at first glance, they in fact exhibit significantly higher diversity and quality, particularly in challenging attributes such as hats or intricate hairstyles (e.g., braids). However, conveying this advantage within the limited space of a teaser is inherently difficult: even methods with low diversity can curate a few visually distinct samples, just as we are constrained to showing only dozens of examples despite our method’s much broader generative capacity.
> To better highlight our improvements under these constraints, we adopt three complementary strategies:
> - Maximized sample number: We pack the teaser and supplementary figures with as many  samples as possible. We encourage readers to compare these directly with sampling results from prior work (via their papers or sampling code) to observe the difference in diversity.
> - Emphasis on rare and complex attributes: We intentionally include examples featuring hats and intricate hairstyles, especially from non-frontal views, which is a capability not achievable by prior works due to their limited diversity and quality. In this revised version, we have also added 360° visualizations of head geometry in the teaser to explicitly showcase the geometry in the back-head region.
> - Beyond visual comparison: Our paper provides deeper analysis and evidence. We provide a detailed analysis in the introduction and Figure 2 of why previous methods are limited in terms of diversity and prone to various artifacts. Moreover, quantitative results in Table 1 confirm our superiority: our method achieves an order-of-magnitude lower FID-random, objectively validating both higher quality and greater diversity.
>
>
> # W2.2 Writing Quality
> We acknowledge using LLMs to assist with language refinement and ensure grammatical correctness. However, we emphasize that all technical contributions and scientific content in this paper are entirely the authors’ own work. We appreciate the reviewer’s feedback and will further improve the clarity and fluency of the writing in the final version.
> Regarding em dash formatting: we followed the style recommended by Webster’s Dictionary, which states: “Most books and journals omit spacing, closing whatever comes before and after the em dash right up next to it.” This convention is also observed in influential papers such as ResNet and Fast R-CNN, which use unspaced em dashes in their abstracts and introductions.
> Nevertheless, we sincerely appreciate the suggestion and have removed all em dashes in the revised manuscript to avoid any ambiguity or stylistic inconsistency.

---

> > ### Author Response · Authors · 2025-11-27
> >
> > # W3: Non-frontal single-view GAN inversion
> > Our method is not limited to strictly frontal images. Specifically, we perform single-view GAN inversion using Pivotal Tuning Inversion (PTI), which optimizes both the latent code $w$ and generator parameters to match the target image via a combination of pixel-wise L2 loss and image-level LPIPS loss. Critically, this inversion process is independent of the semantic condition $c$, i.e., it does not involve predicting or optimizing $c$. Therefore, like prior methods, our approach can, in principle, handle target images from any viewpoint.
> >
> > However, a practical limitation arises from head pose estimation: for large pose variations (e.g., profile views and back views), estimated poses are often inaccurate. And PTI relies on an accurate head pose to align the rendered output with the target image. For this reason, frontal views are typically used to evaluate inversion performance, to avoid confounding factors from pose estimation errors. We also include additional inversion results on non-frontal images in Figure 14 to demonstrate real-world applicability.
> >
> > # Q1: Conditioning vs. Dataset Improvements
> > Using the proposed view-invariant semantic conditioning plays a key role in our method and is our most important core contribution. Without this semantic conditioning, even when trained on our new dataset, the generator still cannot solve the directional bias and limited diversity issues. The new FLUX-generated dataset is designed to facilitate the semantic conditioning, rather than to be used independently. Therefore, we highlight the new conditioning mechanism rather than the new dataset in our title.
> > To validate the necessity of this semantic conditioning, we trained two ablation baselines on the same FLUX-generated dataset:
> > 1. Conditioning only on view (i.e., camera parameters), as in previous methods. This baseline aims to demonstrate that, even when trained on the new dataset, conventional view-based conditioning cannot resolve the directional bias and limited diversity issues.
> > 2. Conditioning on both view and semantic features. This baseline aims to show that removing view information from the conditioning is necessary to achieve view-invariant generation.
> >
> > As shown in Figure 5(f,g), both baselines exhibit strong view-dependent bias: they produce realistic results at the conditioned view but suffer from noticeable distortions at other views. Table 1 quantifies this trend: both achieve low FID at the conditioned view (FID-view) but significantly higher FID on random views (FID-random).
> > These results confirm that even with the improved dataset, the view-invariant semantic condition is essential for consistent, high-quality 3D-aware generation across all viewpoints.

---

> > > ### Author Response · Authors · 2025-11-27
> > >
> > > # Q1.1: Influence of dataset size and number of views
> > > For dataset size, please refer to our response to W1 or to Supp. A.6.1.
> > >
> > > Below is our discussion on the number of views in the dataset.
> > > Our method does not require a strictly multi-view dataset. The primary purpose of our data collection is to obtain front-view image CLIP features as the condition to train our view-invariant semantic-conditioned 3D-aware GAN. In fact, only a single paired sample per identity is minimally required, because any non-frontal view along with its corresponding frontal image is sufficient for training.
> > > In practice, however, we generate multiple views per frontal image for two key reasons:
> > > 1. Computational efficiency: Generating a frontal image satisfying our selection criteria is significantly more expensive than generating non-frontal views. As described in Section 3.1, we need to select the best frontal candidate from multiple generations, making one frontal image cost several times more than a single non-frontal view. To maximize data utility under limited compute, we generate as many non-frontal views as possible per valid frontal image, rather than just one.
> > > 2. Distributional consistency: Using multiple views per identity ensures that the data distribution (e.g., hairstyle, ethnicity, expression) remains balanced across all viewpoints. For example, in an extreme case where short hair dominates the left views while long hair dominates the right views, the model may produce implausible “chimeric” heads, i.e. , short on one side and long on the other, which is also mentioned in SphereHead. Generating multiple views per identity mitigates this risk, as all views preserve the same identity-related attribute distribution as the frontal view.
> > >
> > > Finally, controlling view generation with FLUX.1-Kontext is inherently noisy. Even when prompting for a specific view (e.g., “left view (yaw=90°)”), the model often outputs images at unintended yaw angles (e.g., 30°, 75°, or even 145°), and occasionally generates completely mismatched views (e.g., front, back or right when “left” was requested). Thus, even if we restrict prompts to fewer directions (front, back, left, right), the actual output still spans a continuous range of poses, making it impractical to enforce a fixed, small set of discrete views.
> > >
> > >
> > > # Q2: Quality of FLUX-Generated Images
> > > We randomly visualize additional examples from the FLUX-generated dataset in Figure 12 of the revised version to better illustrate its quality and diversity. As shown, FLUX.1-Kontext generates non-frontal views that preserve key personal attributes from the frontal image, such as identity, expression, hairstyle, and lighting. As discussed in Section 3.2, while these images are not strictly multi-view renderings with perfect 3D consistency, they are sufficient to be compatible with our training pipeline, which ultimately produces 3D full-heads with strong 3D consistency.
> > >
> > > # Q3: Necessity of View-invariant Conditioning
> > > Yes, the view-invariant semantic conditioning is the key innovation that enables our method to succeed. Even when trained on our new dataset, existing approaches still suffer from the directional bias illustrated in Figure 2(d–i). In contrast, adopting our view-invariant semantic condition completely eliminates this issue, which is a technical contribution far from trivial. We therefore consider the analysis and design of the conditioning mechanism to be the core contribution of this work, which is why it is highlighted in the title. Please see our response to Q1 or to Supp. A.6.3 for further details.

---

> > > > ### Author Response · Authors · 2025-11-27
> > > >
> > > > # Q3.1 Clarification of “Align” and “Anchor”
> > > > In our context, “align” and “anchor” refer to the process of using the frontal view corresponding to a given non-frontal image as the source for semantic conditioning.
> > > > The logic is:
> > > > - We need a condition to facilitate training, or it will suffer from mode collapse, Fig. 2(a–c).
> > > > - However, traditional view conditioning leads to the directional bias issue Fig. 2(d–i).
> > > > - So, we need a view-invariant condition.
> > > > - It is inherently difficult to extract a truly view-invariant feature from a single image, as the content of any image inherently encodes its viewing perspective.
> > > > - We use (align/anchor) the random view image’s corresponding frontal-view image to extract the feature as the semantic feature.
> > > > - Since all views derived from the same frontal image share the same semantic feature, this feature is de facto view-invariant.
> > > > - Experiments prove that using such a view-invariant semantic feature as the condition effectively prevents mode collapse and solves the directional bias issue.
> > > >
> > > > Supp. A.6.6 and A.6.7 further justify why the frontal view is the optimal choice for anchoring, due to its richness in identity-critical information and stability under our data generation pipeline.
> > > >
> > > > # Q3.2 Training and Inference Requirements for Semantic Conditioning
> > > > During inference, our method requires only a single input image. We use PTI for single-view inversion and do not predict or optimize the semantic condition c. Therefore, our method is not restricted to frontal views at inference time. Please see Supp. A.6.2 and Figure 14 for details.
> > > >
> > > > It should be noted that during training, we treat all data as individual single images, without any explicit multi-view supervision. This design allows our method to tolerate minor 3D inconsistencies present in the multi-view dataset (see Section 3.2).
> > > >
> > > > In principle, if one had a reliable model to predict the frontal-view CLIP feature from arbitrary views, a multi-view dataset would not be necessary. However, training such a predictor itself requires multi-view data precisely to learn the mapping from non-frontal to frontal semantics. Thus, multi-view data remains essential at the training stage, even if not used directly during inference.
> > > >
> > > > # Q4 Non-frontal single-view GAN inversion
> > > > Please refer to Supp. A.6.2 and Figure 14 for details.
> > > >
> > > > # Q5: Generalizability to Other 3D-Aware Generators
> > > > Yes, all results presented in the paper, including ablation baselines, are based on HyPlaneHead. However, our proposed view-invariant semantic conditioning is compatible with other 3D-aware generator architectures, such as tri-plane (EG3D), tri-grid (PanoHead), and single or dual spherical tri-plane (SphereHead), as they all share the same StyleGAN2-based training pipeline, the only difference being their 3D representation.
> > > > We include a detailed comparison in Supp. A.6.9. The results show consistent trends across architectures: applying our semantic conditioning effectively eliminates directional bias (similar to Fig. 2(d–i)) and yields significantly more diverse outputs.
> > > >
> > > > Notably, symmetry artifacts are largely reduced, though they still occasionally appear in tri-plane and tri-grid representations. Additionally, though at a lower frequency, multiple-face artifacts persist across all representations, including HyPlane.
> > > > We further validate that incorporating the ViCiCo loss quickly resolves the multiple-face artifacts (typically within a few hundred kimg), consistent with prior work.
> > > >
> > > > In summary, our semantic conditioning generalizes well to various 3D generator backbones, consistently improving output quality and diversity. Our main experiments use the most advanced representation (HyPlaneHead), which achieves the best overall performance.
> > > >
> > > > Please refer to Supp. A.6.9 and Figure 16 for details.
> > > >
> > > > # Q6: GPU Type Clarification
> > > > We use NVIDIA A10 GPUs (24 GB VRAM, Ampere architecture) to generate our dataset. The specific model we employ is FLUX.1-Kontext-dev(https://huggingface.co/bullerwins/FLUX.1-Kontext-dev-GGUF/blob/main/flux1-kontext-dev-Q8_0.gguf), which is able to runs efficiently within the 24 GB memory limit after 8-bit quantization.
> > > >
> > > > # Q7: Dataset Release
> > > > The dataset, along with the code for data generation and preprocessing, will be made publicly available upon acceptance of this paper.

---

> > > > > ### Author Response · Authors · 2025-11-27
> > > > >
> > > > > # Q8: Clarification on View Selection for Semantic Conditioning
> > > > > We apologize for the confusing wording in the original manuscript. This has been revised for clarity in the updated version.
> > > > >
> > > > > Our intended point is this: no single view captures 100% of the full-head semantic information. However, among all viewpoints, the frontal view provides the most comprehensive signal. It fully captures the facial region, which is most critical to human perception, and also conveys substantial global cues about hairstyle, hair color, clothing, and overall appearance.
> > > > >
> > > > > For example, if the frontal view shows a red afro hairstyle, the back view is highly likely to show the same red afro, but not a green ponytail. In other words, while the frontal view may miss some fine details (e.g., back-of-head hair geometry), it still encodes the majority of identity-relevant semantics, whereas side or rear views capture significantly less, often lacking facial identity entirely.
> > > > >
> > > > > Therefore, we use the frontal view as the source of semantic conditioning, not because it is perfect, but because it is the most informative single view available.
> > > > >
> > > > > # Q9: Why Not Fuse Semantic Information from Multiple Views?
> > > > > We do not fuse semantic conditions from multiple views not because of inference constraints (PTI optimizes only the latent code $w$, without predicting or optimizing the condition $c$), but for the following key reasons:
> > > > >
> > > > > 1. Frontal view quality is significantly more reliable. Our data pipeline generates multiple frontal candidates and selects the one with the most accurate pose, highest visual quality, and strongest identity alignment with the original real image (see Figure 3). In contrast, non-frontal views suffer from unstable pose control (e.g., requesting a left profile sometimes yields yaw = 45° instead of 90°) and occasional identity drift or artifacts, despite post-generation filtering. Thus, non-frontal views are less trustworthy as semantic sources.
> > > > > 2. Semantic consistency across views is anchored to the frontal image. All non-frontal views are generated from the selected frontal image using view prompts. This ensures minimal semantic drift between views. Using multiple independent views as conditions would introduce random inconsistencies (e.g., hairstyle, identity, expression mismatches), leading to conflicting semantic signals and unstable conditioning.
> > > > > 3. Computational and architectural overhead. Our generator uses a StyleGAN2 backbone, where the condition $c$ (a 512-dimension CLIP feature) and the random number $z$ are projected into latent space $W$ via an MLP. Concatenating features from multiple views would drastically increase both parameter count and computational cost, with diminishing returns.
> > > > > 4. The frontal view is already the most informative single view. As explained in our response to Section A6.6, while no single view captures 100\% of full-head semantics, the frontal view contains the majority of identity-critical information, including face, hairstyle, color, and global appearance, which makes it the optimal choice for semantic conditioning.
> > > > >
> > > > > For these reasons, we use only the frontal view to extract the semantic condition. Content in Section 4.2 is also informative for this discussion.

---

### Author Response · Authors · 2025-11-27
**Official Comment by Authors**

We sincerely thank the reviewers for their valuable suggestions, which have greatly helped us improve the paper. In this revision, we have addressed the feedback as follows (newly added content is highlighted in blue in the manuscript):

• Figure 1 has been updated to include more diverse generated samples and multi-view geometry visualizations, better highlighting our method’s advantages in both diversity and fidelity.

• Figure 2 has been corrected to fix minor typos.

• We have revised the wording of several sentences throughout the paper to improve clarity, following the reviewers’ writing suggestions.

• As suggested, we have added new quantitative comparisons with state-of-the-art methods, including Mean k-Nearest Neighbor Distance (MKNND) for identity diversity and LPIPS for reconstruction fidelity (see Table 2).

• Three additional comparison videos showing baseline results under random-view conditioning have been included in the supplementary material.

• A new Discussion section has been added to the supplementary material, summarizing key points raised by the reviewers and our corresponding responses.

• Figure 13 now includes a visualization of samples from the training dataset.

• Figure 14 presents single-view GAN inversion results using both near-frontal and non-frontal views.

• Figure 16 demonstrates our method’s compatibility with other tri-plane-like representations.

• Figure 17 shows a nearest-neighbor analysis comparing generated samples to their closest matches in the training set.

---

### Author Response · Authors · 2025-11-28
**Response to the Core Concerns**

We summarize each reviewer’s most central concern below and provide a concise response.

---

**WZgK: Can the method be inverted from non-strictly-frontal single images?**

Yes. Our generator uses a StyleGAN2 backbone where the latent code $w$ is derived from both the random vector $z$ and condition $c$. Since inversion (via PTI) directly optimizes $w$ without requiring estimation of $c$, our approach works with arbitrary views, not just strictly frontal ones. See Figure 14 for non-frontal inversion results.

---

**t6Lb: Will the curated dataset be released?**

Yes. The dataset, along with code for data generation and preprocessing, will be publicly released upon acceptance.

---

**9wYw: Need more quantitative results.**

In addition to the FID metrics in Table 1, we now report MKNND (for identity diversity) and LPIPS (for reconstruction fidelity) in Table 2. Our method achieves state-of-the-art performance on both.

---

**VqNs: The dataset is quite valuable, but the technical contribution is incremental.**

To clearly clarify our contributions and encourage further discussion, this response is somewhat lengthy.

We appreciate this comment. However, we would like to emphasize that our primary contribution lies not in the dataset itself, but in the analysis and innovation of view-invariant semantic conditioning. This is why our title highlights conditioning rather than the dataset. The dataset was created specifically to support the training of our semantic-conditioned GAN, and we demonstrate its value through three key aspects below.

**1. Analysis:**

Directional bias (illustrated in Figure 2) is a well-known problem that has hindered 3D-aware GANs for years. This issue leads to visual artifacts and limits models’ ability to achieve full diversity and high visual quality. Many prior works, including EG3D, PanoHead, SphereHead, HyPlaneHead, and GGHead, have acknowledged this problem in their papers and technical forums (e.g., GitHub issues). While they have attempted to mitigate it through various strategies such as regularization or alternative representations, none have fully resolved it.

To the best of our knowledge, this paper is the first to systematically analyze directional bias and identify its root cause: using viewpoint as a conditioning signal. Furthermore, we propose a promising direction to address it: replacing view-dependent conditions with a view-invariant semantic condition.

**2. Solution:**

We propose using the front-view CLIP feature as a view-invariant condition, which is simple yet highly effective. To enable training with this strategy, we also introduce a new dataset. Together, these technical contributions allow our method to eliminate directional bias at its source, resulting in significantly higher diversity, fewer artifacts, and consistent improvements when integrated into existing architectures.

**3. Broader Impact:**

As the first photorealistic 3D full-head model trained on over 10 million images, our method demonstrates strong diversity and generalizability. It's potential to serve not only as a foundational 3D head model for applications like 3D talking heads or 3D head editing, but also as a high-quality 2D/3D photorealistic head data generator for broader downstream tasks such as 3D reconstruction.

Moreover, our work successfully leverages imperfect, 3D-inconsistent multi-view generated data to train a model that produces fully 3D-consistent heads. We hope this work inspires the community to rethink how 2D data can be leveraged for training 3D models. Rather than relying exclusively on either strictly consistent multi-view data (expensive) or purely single-view 2D data (inefficient and introduces view-imbalance), there exists a promising intersection: imperfect multi-view data generated from powerful 2D generators.

The key insight is that research should not focus solely on generating more 3D-consistent multi-view images, but also on developing inconsistency-tolerant 3D models or training strategies that can effectively learn from imperfect multi-view data while being robust to minor inconsistencies. Our method represents an initial exploration in this direction, highlighting that view-invariant semantic information is crucial to bridging this gap.

---

### Meta-Review · Area_Chair_kKSN · 2026-01-01

**Summary:**

The paper present a modification to 3D GAN approach for head generation, there are 2 main contribution behind this paper:
1) Large scale dataset constructed with the help of Flux Kontext.
2) The new method of conditioning based on semantic CLIP features extracted from frontal view image.

All the concerns regarding comparison to baselines and evaluation of the inversion was addressed during rebuttal phase. However area chair has concerns of his own see below, but area chair believe they could be addressed in the final version.

**Reviewer Concerns:**

1) **Unclear comparison with state-of-the-art methods.** Because the proposed method required a specific dataset (where there is exists frontal view image for each identity) it is hard to compare with pre-trained methods. Authors re-train baseline methods on their dataset and compare, which is a reasonable solution.

*It is still however unclear from the manuscript itself which row in the Table 1 and Table 3 corresponds to which, state-of-the-art method.*

*Moreover it seems, one point overlooked by the reviewers, but crucial for fair evaluation is how inference is performed, mainly from where view-invariant semantic conditioning is taken from? If it comes from same gt data used to compare FID against, this will make it closer to rFID (reconstruction FID) rendering comparison unfair.*

2) **Dataset release.** The authors promised to release the dataset.

3) **Inversion**. Inversion from non-frontal images seems to be fairly addressed by 14. However, Table 2 is still lacking evaluation of the head geometry.

4) **Technical contribution/novelty**. Seems adequate.

5) **Poor video presentation in supplementary**.

*This is still an issue, there are more videos, but it hard to compare proposed method vs baselines. For final version, it is better to combine videos in website with proper explanations.*

**Reviewer Scores:**

WZgK, will likely increase to 5.

t6Lb, will stay the same.

9wYw, will stay the same.

VqNs, will stay the same.

---

### Decision · Program_Chairs · 2026-01-26

Accept (Poster)